# Deformba: Vision State Space Model with Adaptive State Fusion

**Hongyu Ke**[1]  **Jack Morris**[2]  **Yongkang Liu**[3]  **Satoshi Kitai**[3]  **Kentaro Oguchi**[3]  **Yi Ding**[2]  **Haoxin Wang**[1]

## Abstract

State Space Models (SSMs) have emerged as a powerful and efficient alternative to Transformers, demonstrating linear-time complexity and exceptional sequence modeling capabilities. However, their application to vision tasks remains challenging. First, existing vision SSMs largely depend on manually designed fixed scanning methods to flatten image patches into sequences, which imposes predefined geometric structures and increases the complexity. Second, the broader adoption of vision SSMs is hindered in domains that require query-based interactions between distinct information streams. This is a result of the inherently causal and self-referential nature of SSMs designed for 1D sequence modeling tasks. This fusion mechanism is indispensable for critical perception tasks such as multi-view 3D fusion. To address these limitations, we propose Deformba, a context adaptive method that dynamically augments the spatial structural information while maintaining the linear complexity of SSMs. Deformba also allows multi-modal fusion like cross attention. To demonstrate the effectiveness and general applicability of Deformba, we test its performance on general 2D vision tasks such as image classification, object detection, and segmentation, as well as 3D vision tasks like BEV perception. Extensive experiments show that Deformba achieves strong performance across various visual perception benchmarks. Code will be available at: https://github.com/amai-gsu/Deformba.git

## 1. Introduction

Recent State Space Models, such as HiPPO (Gu et al., 2020), LSSL (Gu et al., 2021b), and S4 (Gu et al., 2021a), have

[1]Department of Computer Science, Georgia State University [2]University of Tennessee Knoxville [3]InfoTech Labs, Toyota Motor North America R&D. Correspondence to: Haoxin Wang <Haoxinwang@gsu.edu>.

*Proceedings of the 43rd International Conference on Machine Learning*, Seoul, South Korea. PMLR 306, 2026. Copyright 2026 by the author(s).

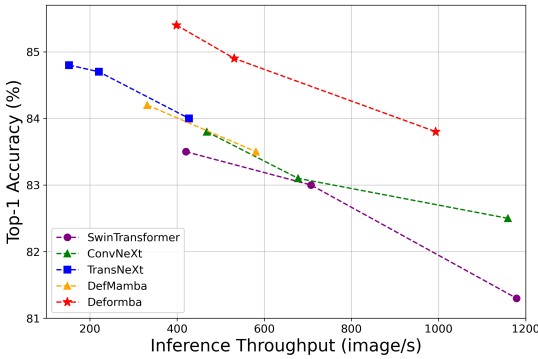

*Figure 1.* The trade-off between ImageNet-1K top-1 accuracy and inference throughput. Actual hardware used for inference throughput is an NVIDIA RTX 6000 Ada GPU with a batch size 128. It can be seen that under the same inference throughput or accuracy, the accuracy or inference throughput of our proposed Deformba outperforms other methods.

gained attention as compelling alternatives to Transformers in domains traditionally dominated by them (Zhu et al., 2026b; Xu et al., 2025a; 2026; Liang et al., 2026a; Zhu et al., 2026a; Liang et al., 2026b; Xu et al., 2025c; 2024; 2025b). By explicitly modeling the evolution of hidden states, they achieve high efficiency in sequence processing. In particular, Mamba (S6) (Gu & Dao, 2023) and Mamba2 (Dao & Gu, 2024) further improve this family with a selective scanning mechanism and hardware-aware parallelism to enable efficient training and inference, achieving comparable performances to Transformers in natural language processing tasks. This motivates SSMs to be applied for 2D vision tasks such as classification, object detection and segmentation (Ma et al., 2025; Lee et al., 2025; Zhu et al., 2024; Liu et al., 2024; Huang et al., 2024; Shi et al., 2025).

Despite this progress, extending SSMs into a practical vision SSM paradigm remains challenging. Standard SSMs are built around a causal and self-referential state evolution on 1D sequence, whereas visual understanding requires non-causal spatial interaction over 2D feature maps. Previous vision SSMs work addresses this mismatch by employing hand-crafted scan orders to flatten 2D features into 1D sequence, such as sweeping scanning (Zhu et al., 2024), continuous scanning (Yang et al., 2024a), and local scanning (Huang et al., 2024) (Figure 2(a)-(c)). While these designs partially improve the alignment between spatial structure and sequential computation, they remain constrained by

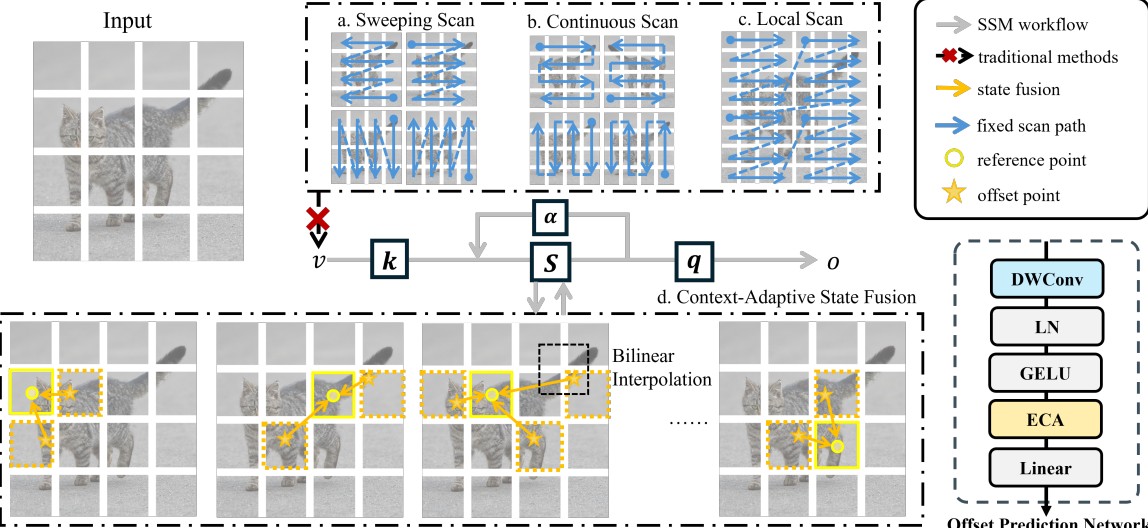

*Figure 2.* **Context-Adaptive State Fusion (CASF).** Instead of relying on multiple predefined scanning methods (e.g., sweeping/continuous/local), CASF decouples the SSM write/read and inserts an offset predictor to sample context-adaptive evidence from the hidden state map and fuse it into the current state without additional scanning.

fixed scanning paths that a small set of scanning direction (e.g., the commonly used four-directional scanning) might be insufficient to capture complex spatial relationships, whereas adding more directions increases computation and memory. Moreover, any fixed scanning path inevitably perturbs the native 2D neighborhood structure. Serialization can break spatial continuity and bias interaction patterns toward the chosen scanning direction, which makes spatial modeling sensitive to the imposed ordering (Xiao et al., 2024a; Liu et al., 2025a).

On the other hand, many modern perception systems rely on query-to-source interactions between distinct feature streams, such as fusing multi-view features or aggregating multi-scale representations. A widely adopted architectural pattern is to maintain a set of task-specific query tokens and let them selectively gather information from one or more sequences of source features. Representative examples include BEV grid queries that aggregate evidence from multi-view image features for 3D perception (Li et al., 2022a; Yang et al., 2023; Ke et al., 2025b), and object queries that capture information from multi-scale feature maps for 2D detection (Zhu et al., 2020; Wang et al., 2022b; Huang et al., 2025). This query-based fusion is central to tasks beyond single-image recognition, yet it is not naturally supported by conventional SSM formulations, but commonly realized by cross-attention in Transformer architectures. This raises a fundamental methodological question: *Can the SSM paradigm be extended into an efficient vision paradigm that is effective not only for general 2D image tasks, but also for query-to-source interaction, while retaining the efficiency advantages of state-based computation?*

Motivated by this perspective, we propose Deformba, a vi-

sion state space framework that extends the vision SSM paradigm in two complementary directions. At its core, Deformba introduces Context-Adaptive State Fusion (CASF), which performs context-adaptive state reading to dynamically gather spatially relevant evidence under the guidance of hidden states. Specifically, a single unidirectional scan is only used to fit into SSM and write a shared state memory. The interaction topology is learned by spatial aware sampling rather than predefined scan methods. As a result, Deformba enables flexible spatial interaction for single-image 2D tasks, and also supports query-to-source fusion where external queries sparsely retrieve spatial evidence from the same shared memory. These capabilities are unified within one design, allowing Deformba to serve both general 2D perception and multi-modal 3D perception.

We conducted extensive experiments to evaluate the effectiveness and general applicability of Deformba. For general 2D vision tasks, we test image classification, object detection, instance segmentation, and semantic segmentation. We further evaluate Deformba on the 3D perception task such as BEV 3D detection where multi-modal fusion is essential. Extensive experiments demonstrate that Deformba achieves or surpasses the performance of state-of-the-art benchmarks.

## 2. Related Work

### 2.1. Vision State Space Models as Alternatives to Self-Attention

SSMs have emerged as a compelling alternative to Transformers. A major advance is Mamba (Gu & Dao, 2023; Dao & Gu, 2024), which introduces the selective state space model (S6). As with other linear attention methods (Schlag

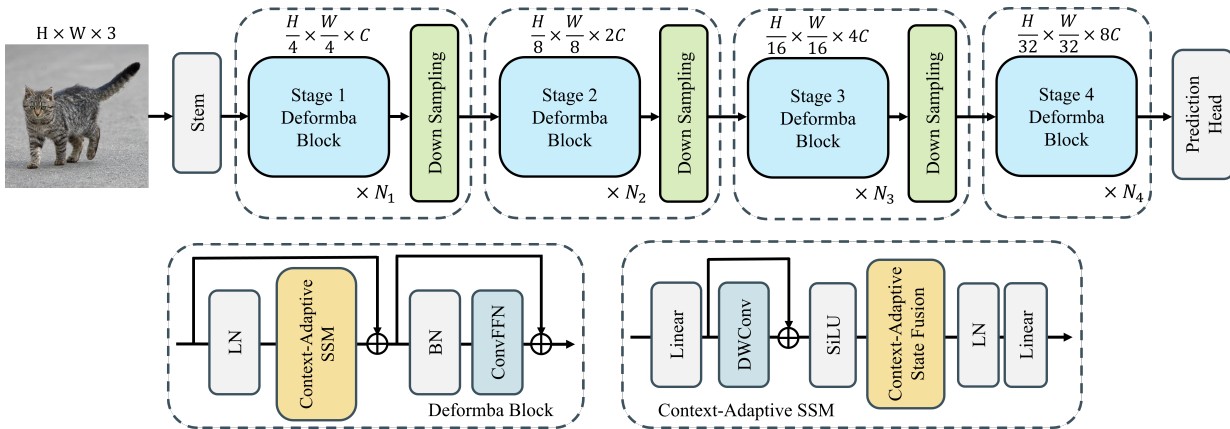

*Figure 3.* **Deformba architecture and block designs.** The network adopts a four-stage hierarchical backbone with downsampling between stages and stacks Deformba blocks; each block consists of a Context-Adaptive SSM (with CASF) and a ConvFFN under residual connections.

et al., 2021; Yang et al., 2024b), Mamba employs input-dependent parameters to enable interaction across sequence elements. However, adapting these inherently 1D models to 2D vision tasks presents challenges. Previous works (Zhu et al., 2024; Liu et al., 2024; Yang et al., 2024a; Huang et al., 2024; Shi et al., 2024; Xiao et al., 2024b) successfully adapt State Space Models to vision by treating spatial dimensions as sequences through different predefined scanning methods. Approaches such as Vision Mamba (Vim) (Zhu et al., 2024) and VMamba (Liu et al., 2024) treat an image as a sequence of flattened patches. To preserve spatial relationships, they often employ a multi-scan strategy. While effective to some extent, processing the data multiple times with different predefined flattening orders (e.g., row-wise and column-wise) increases the complexity and introduces architectural rigidity, highlighting an unnatural fit between the causal structure of SSMs and the non-causal nature of spatial data.

## 2.2. Vision State Space Models as Alternatives to Cross-Attention

Most existing SSM-based vision models (Xiao et al., 2024a; Ma et al., 2025; Liu et al., 2025a; Li et al., 2025) still operate on a single image or feature modality. They do not provide a general mechanism for query-based fusion across multiple feature sequences, such as multi-view, multi-scale, or multi-modal streams. In particular, there is no SSM counterpart to cross-attention. The causal state evolution in standard SSMs makes it difficult for a query sequence to non-causally read from other sequences. In contrast, our work explicitly targets this gap by endowing SSMs with the CASF mechanism between multiple feature streams. The application of SSMs to query-based fusion is a nascent but growing field. One of the examples is BEV perception. Recent works like Voxel Mamba (Zhang et al., 2024), MamBEV (Ke et al., 2025b) and TinyBEV (Ke et al., 2025a) represent important first steps. Voxel Mamba introduces a group-free strategy that

serializes the entire voxel space into a single sequence by employing a dual-scale SSM block. MamBEV and Tiny-BEV replace the standard cross-attention in Transformer-based models with an SSM-based cross-attention module, retaining the separate query-and-pull fusion paradigm but with linear complexity. These methods primarily use SSMs as a component replacement within existing architectural frameworks. And they concatenate or mix queries and multi-direction image features into a single sequence, passing to SSM equation directly. In contrast, our proposed Deformba allows the model to break free from the rigid causality of the SSM scan, enabling it to aggregate features from the entire global context without resorting to quadratic-cost attention or inefficient multi-scan patterns.

## 3. Preliminaries

**State Equation**. The Mamba SSM maintains a hidden state $\mathbf{S}_t$ to compress information in the sequence, which is updated recurrently as follows:

$$\mathbf{S}_t = \alpha_t \mathbf{S}_{t-1} + \boldsymbol{v}_t \boldsymbol{k}_t^\top \in \mathbb{R}^{d_v \times d_k}, \tag{1}$$

where $\boldsymbol{v}_t$, $\boldsymbol{k}_t$, and $\alpha_t$ are a function of the input token $x_t \in \mathcal{X}$. In a classic construction of linear attention, the input sequence decodes the final state $\mathbf{S}_T$ using the following function:

$$\mathbf{O} = \mathbf{S}_T \mathbf{Q} \in \mathbb{R}^{L_q \times d_v} \tag{2}$$

In Mamba and Mamba2, the state equation compresses the key-value sequence using a decay term $\alpha_t \in \mathbb{R}^{d_\alpha}$, which acts as a forget gate for information which is no longer relevant to the current token. In self-attention settings, The query, key, and value sequences are based on the input sequence, where the relevance of stored information can be inferred directly from the current token.

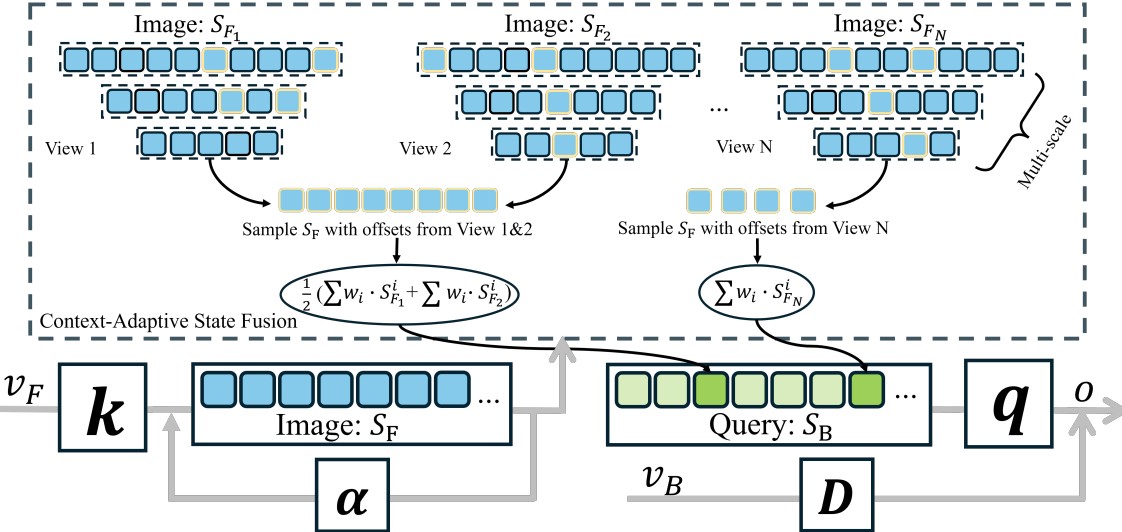

*Figure 4.* Deformba Cross Attention. To enhance spatial understanding under Mamba's causal constraints, each query samples relevant image features using learned offsets. This process enables global spatial interaction without expensive scans.

When emulating cross-attention, the state equation above is used to compress information from the key and value input, and the query input sequence decodes the final state $\mathbf{S}_T$. However, in cross-attention settings, the relevance of the information in the state is a function of the query input sequence. If all information in the sequence cannot be compressed into a state of a given size, then a larger state size is required. For example, BEV encoding has a static key-value sequence length, but the information density varies. Additionally, the downstream tasks, such as BEV, demand a highly accurate representation of the environment, meaning the state should be large enough to accommodate edge cases but also requires high efficiency for edge deployment. These requirements would naively push SSM-based cross-attention towards ever larger state sizes, undermining its efficiency advantages. Instead of increasing the capacity of $\mathbf{S}_t$, we seek a mechanism that keeps the recurrent state compact and hardware-friendly, yet allows queries to selectively access information from the key–value sequence.

## 4. Methods

Our target is to update each image feature/query using relevant information in the sequence. In this section, we first describe the overall architecture of the proposed network for 2D image tasks. Next, we introduce Context-Adaptive State Fusion (CASF). Then, we describe how the Deformba block is adapted for cross-attention. Finally, we provide our theoretical analysis of the computation.

### 4.1. Network Architecture.

The overall architecture of Deformba is shown in Figure 3. The input image $x \in \mathbb{R}^{H \times W \times 3}$ passes through an overlapped stem layer to generate 2D feature map of size $\frac{H}{4} \times \frac{W}{4} \times C$. Then this feature map is fed into four successive stages to create hierarchical representations of dimensions $\frac{H}{8} \times \frac{W}{8} \times 2C$, $\frac{H}{16} \times \frac{W}{16} \times 4C$ and $\frac{H}{32} \times \frac{W}{32} \times 8C$. Each stage is composed of multiple stacked Deformba blocks followed by a downsampling layer by a factor of 2 and feature dimension increase, except for the last stage. Finally, a prediction head is employed to process these features. Specifically, the Deformba block serves as the fundamental building block of our architecture, comprising a Context-Adaptive SSM for state fusion and a convolutional FFN with skip connections to increase locality inductive bias.

### 4.2. Context-Adaptive State Fusion

As shown in Figure 2, we propose CASF, which is designed to address the limitation of predefined scanning methods in Vision State Space Models. Existing approaches restrict spatial interaction to fixed scan orders. In contrast, CASF enables each image feature to adaptively aggregate spatially relevant information under the guidance of its hidden state. The core mechanism of our method splits the SSM computation into read and write steps and inserts an offset prediction network to obtain the positions of the focused regions in the middle. We decouple the SSM write operation from the SSM read operation, applying them to image features.

Specifically, we first leverage the SSM's write recurrence from Eq. 1 to encode the entire 2D image feature map $\mathbf{F}$. This operation, performed using a single sweeping scan, generates a context-aware hidden state sequence $\{\mathbf{S}_t\}_{t=1}^{L}$ and its 2D form $\mathbf{S}_{2D} \in \mathbb{R}^{H \times W \times C}$. Based on $\mathbf{S}_{2D}$, we predict a set of sampling offsets to retrieve spatial evidence in a data-driven manner, instead of relying on predefined neighborhood patterns.

Formally, given the states $\{\mathbf{S}_t\}_{t=1}^{L}$ from the SSM write pass,

we predict spatial offsets for each token to retrieve informative evidence from the 2D state map $\mathbf{S}_{2D}$. Specifically, we employ a lightweight offset prediction network $f_{\text{off}}(\cdot)$ to generate $G$ groups of 2D offsets:

$$\Delta p = f_{\text{off}}(\mathbf{S}_{2D}) \in \mathbb{R}^{H \times W \times 2G}, \quad (3)$$

where $G$ is defined as $\frac{d_{model}}{d_{head}}$, in our setting, $G = 8$.

Based on previous work that Mamba-based architectures inherently suffer from channel redundancy (Shaker et al., 2025; Ma et al., 2025), and that offset prediction benefits from global perception. We apply Efficient Channel Attention (ECA) (Wang et al., 2020) after a depthwise convolution in $f_{\text{off}}(\cdot)$ to suppress redundant channels and inject global context through global average pooling with negligible overhead. $\text{ECA}(\cdot)$ can be expressed as:

$$\text{ECA}(\mathbf{U}) = \mathbf{U} \odot \sigma\big(\text{Conv1D}\big(\text{Pool}_{H,W}(\mathbf{U})\big)\big), \quad (4)$$

where $\mathbf{U} \in \mathbb{R}^{B \times C \times H \times W}$, $\text{Pool}_{H,W}(\cdot)$ denotes average pooling over spatial dimensions, i.e., $[\text{Pool}_{H,W}(\mathbf{U})]_{b,c} = \frac{1}{HW} \sum_{i=1}^{H} \sum_{j=1}^{W} U_{b,c,i,j}$, $\text{Conv1D}(\cdot)$ is applied along the channel dimension, and $\sigma(\cdot)$ is the sigmoid function.

Given the reference point coordinates $\mathcal{E}$, which is centered on the image feature query in self attention case. For cross attention in BEV perception case, the reference point $\mathcal{E}$ follows BEVFormer style initialization, the sampling positions are obtained by:

$$\mathcal{P} = \mathcal{E} + \Delta p. \quad (5)$$

Since $\mathcal{P}$ contains a decimal part, it cannot be used directly. We use bilinear interpolation to sample features from $\mathbf{S}_{2D}$ at these positions and fuse them back to update the current state along with learned weights $w \in \mathbb{R}^{H \times W \times G}$:

$$\mathbf{S}_Q = \sum_{g=1}^{G} w_g \cdot \phi(\mathbf{S}_{2D}, \mathcal{P}_g), \quad (6)$$

where $\phi(\mathbf{S}_{2D}, \mathcal{P}_g)$ denotes bilinear sampling on $\mathbf{S}_{2D}$ at location $\mathcal{P}_g$. This spatial-aware sampling process efficiently captures information across the global receptive field without additional expensive scans. $w$ is generated from processing $\mathbf{S}_{2D}$ with a linear layer. In practice, the number of scans is 1 and $|\Delta p|$ is small, leading to an efficient complexity of $O(L)$. The final output is generated from the enriched state $\mathbf{S}_Q$ using the output projection equation:

$$\mathbf{O} = \mathbf{S}_Q \mathbf{Q} + \mathbf{V} \mathbf{D}, \quad (7)$$

where $Q$ and $D$ are the projection parameters associated with $V$. The remaining operations follow that of Mamba, but we omit the element-wise gating, retaining the layer norm followed by a linear layer.

Therefore, global long-range dependencies with spatial details are injected into the state map and augmented representational capacity. This enhances the model's adaptability

and comprehensive scene understanding. Notably, different from DefMamba, which introduces deformability into the scanning process itself by predicting both 2D point offsets and token-index offsets, our Deformba keeps the SSM write pass fixed. Instead of dynamically modifying the scan order before state writing, Deformba introduces adaptivity during the read stage.

### 4.3. Deformba Adapted for Cross-Attention

A key advantage of Deformba is that its *write–read decoupled* formulation is not tied to self-attention. The proposed CASF first *writes* a context-aware state memory from a source sequence using a single SSM scan and then *reads* spatial evidence from the written state map through learned sampling offsets. This paradigm makes it directly applicable to cross-attention. The general problem setup is we aim to take a set of input image feature maps $\mathbf{F}$ and a set of queries $\mathbf{B}$. The module is tasked to select information from $\mathbf{F}$ to update $\mathbf{B}$ in a supervised learning problem. An illustration is displayed in Figure 4 for the multi-view case.

**Minimal changes.** Compared to Deformba self-attention block, the SSM *write* pass is unchanged: we still scan the source image tokens once to obtain a context-aware state memory. The only difference lies in the *read* pass: an external set of queries $\mathbf{B}$ produces the query embedding $\mathbf{Q}$ and predicts query-specific offsets, so each query samples a distinct set of locations from the shared state map and aggregates them into $\mathbf{S}_Q$ to update the final output.

With these changes, Deformba-XA retains the same computational pattern as Deformba. Therefore, it shows the general applicability of Deformba that supports both self-attention style feature mixing and cross-attention style query-to-context aggregation with the same underlying CASF mechanism. More details about Deformba Cross Attention are provided in the Appendix section A.1.

Our CASF decouples write and read, therefore the same mechanism supports both self-attention-style mixing and cross-attention-style query-to-source retrieval with minimal changes to the write pass. This write-once, adaptive-read-many paradigm is the key conceptual novelty of our work beyond simply adding a deformable module to Mamba.

### 4.4. Efficiency

Selective State Space models and more generally linear attention models typically use one of two algorithms to leverage powerful parallelization hardware during training. The first is an associative parallel scan, which can compute the states and outputs for a sequence of length $N$ in $O(N)$ total work. The second is a blockwise associative parallel scan, which computes the states and outputs in chunks, leveraging modern SIMT GPU compute architectures more

| Classification results on ImageNet with 300 epochs | | | | |
|---|---|---|---|---|
| Method | Type | Params(M)↓ | FLOPs(G)↓ | Top-1(%)↑ |
| ConvNeXt-T | CNNs | 29 | 5.0 | 82.5 |
| RegNetY-4G | CNNs | 21 | 4.0 | 80.0 |
| MambaOut-T | CNNs | 27 | 4.5 | 82.7 |
| UniRepLKNet-T | CNNs | 31 | 4.9 | 83.2 |
| InternImage-T | CNNs | 30 | 5.0 | 83.5 |
| SwinV2-T | ViTs | 28 | 4.4 | 81.8 |
| NAT-T | ViTs | 28 | 4.3 | 83.5 |
| StructViT-S-8-1 | ViTs | 24 | 5.4 | 83.3 |
| QFormer$_h$-T | ViTs | 29 | 4.6 | 82.5 |
| Swin-T | ViTs | 28 | 4.5 | 81.3 |
| CMT-S | ViTs | 25 | 4.0 | 83.5 |
| VMamba-T | SSMs | 22 | 5.6 | 82.6 |
| MSVMamba-T | SSMs | 33 | 4.9 | 83.0 |
| LocalVMamba-T | SSMs | 26 | 5.7 | 82.7 |
| DefMamba-S | SSMs | 32 | 4.8 | 83.5 |
| Deformba-T | SSMs | 25 | 4.8 | **83.8** |
| ConvNeXt-S | CNNs | 50 | 8.7 | 83.1 |
| RegNetY-8G | CNNs | 39 | 8.0 | 81.7 |
| MambaOut-S | CNNs | 48 | 9.0 | 84.1 |
| UniRepLKNet-S | CNNs | 56 | 9.1 | 83.9 |
| InternImage-S | CNNs | 50 | 8.0 | 84.2 |
| SwinV2-S | ViTs | 49 | 8.5 | 83.8 |
| NAT-S | ViTs | 51 | 7.8 | 83.0 |
| CrossFormer-B | ViTs | 52 | 9.2 | 83.4 |
| QFormer$_h$-S | ViTs | 51 | 8.9 | 84.0 |
| Swin-S | ViTs | 50 | 8.7 | 83.0 |
| TransNeXt-S | ViTs | 50 | 10.3 | 84.7 |
| VMamba-S | SSMs | 44 | 11.2 | 83.6 |
| MSVMamba-S | SSMs | 50 | 8.8 | 84.1 |
| LocalVMamba-S | SSMs | 50 | 11.4 | 83.7 |
| DefMamba-B | SSMs | 51 | 8.5 | 84.2 |
| Deformba-S | SSMs | 45 | 10.3 | **84.9** |
| ConvNeXt-B | CNNs | 89 | 15.4 | 83.8 |
| RegNetY-16G | CNNs | 84 | 16.0 | 82.9 |
| MambaOut-B | CNNs | 85 | 15.8 | 84.2 |
| InternImage-B | CNNs | 97 | 16.0 | 84.9 |
| SwinV2-B | ViTs | 88 | 15.1 | 84.6 |
| NAT-B | ViTs | 90 | 13.7 | 84.3 |
| QFormer$_h$-B | ViTs | 90 | 15.7 | 84.1 |
| Swin-B | ViTs | 88 | 15.4 | 83.5 |
| TransNeXt-B | ViTs | 90 | 18.4 | 84.8 |
| VMamba-B | SSMs | 75 | 18.0 | 83.9 |
| MSVMamba-B | SSMs | 91 | 16.3 | 84.4 |
| Deformba-B | SSMs | 85 | 16.3 | **85.4** |

*Table 1.* Comparison of classification performance on the ImageNet-1K dataset with 300 epochs training.

effectively to reduce runtime, though still in $O(N)$ total work (if $Q$ is chunksize then total work is $O(QN)$). Deformba is able to utilize either method as we set the state size to 1, though we utilize the non-blockwise algorithm in our sample implementation.

**Computation.** The computational complexity of the Deformba Block is dominated by the linear projections and the offset prediction method, both of which scale linearly with the sequence length $L = H \times W$. Unlike standard self-attention mechanisms that require $O(L^2)$ operations to compute pairwise token interactions, the recurrent state update processes the sequence in $O(LCN)$ operations, where $N$ is a fixed state size; hence the complexity is linear in $L$ with $N$ acting as a constant factor. Additionally, the adaptive sampling step introduces a small spatial retrieval overhead that remains efficient due to its sparsity. Instead of attending to all $L$ pixels, each token samples a small, fixed number of points $K$, resulting in a complexity of $O(LK)$ for the sampling step. The Efficient Channel Attention

mechanism used to generate these offsets utilizes a 1D convolution with a fixed kernel size, adding a negligible $O(C)$ computational cost and avoiding the quadratic complexity typically associated with generating dense attention maps for offset regression. Analysis for I/O and memory shows in the Appendix section B.

**Single Scan.** In Deformba, the scan is used to construct a state memory $S_{2D}$, while the final interaction topology is established by CASF through adaptive reading. Formally, the write recurrence $S_t = \alpha S_{t-1} + v_t k_t^T$, defines a fixed directed write graph, whereas CASF introduces learned read edges through Equations 5 and 6. Hence, the effective interaction graph is $G_{eff} = G_{write} \cup G_{read}$, which is not determined by scan order alone. This explains why a single scan can be adequate in practice. The final receptive field of tokens become a union of multiple written state receptive fields selected by adaptive reading. In perspective of state compression, this avoids forcing the write process alone to preserve all 2D dependencies, which would otherwise require a larger state. In this sense, the role of the single scan is efficiency-preserving state construction, while CASF provides the missing non-causal spatial information flow.

## 5. Experiments

### 5.1. Image Classfication

**Settings.** Image classification experiments are conducted on the ImageNet-1K dataset (Deng et al., 2009). Our training setup adopts the commonly used experimental setting in pervious work (Touvron et al., 2021; Liu et al., 2021). The experiments are conducted on 8 A6000 GPUs for 300 epochs on images with a resolution of $224 \times 224$ and a learning rate of $5 \times 10^{-4}$. We used a linear warmup for the first 20 epochs from $5 \times 10^{-7}$. Following the warmup, the learning rate follows a cosine decay schedule with a decrease from the base learning rate to a minimum learning rate of $5 \times 10^{-6}$ with a weight decay rate of 0.1. Optimization is performed with AdamW with betas set to (0.9, 0.999) and momentum at 0.9. To further refine model accuracy and generalization, we incorporate exponential moving average (EMA) techniques.

**Results.** Table 1 presents our proposed Deformba with various state-of-the-art (SOTA) methods such as convolution-based, transformer-based and mamba-based architectures. Specifically, our base model, Deformba-B, achieves 85.4% Top-1 accuracy, which outperforms CNNs based MambaOut-B (Yu & Wang, 2025), ViTs based TransNeXt-B (Shi, 2024) and SSMs based VMamba-B by 1.2%, 0.6% and 1.5%, respectively, with similar amount of parameters and FLOPs. Compared to models categorized as Small and Tiny, Deformba consistently outperforms its counterparts by achieving Top-1 accuracies of 84.9% and

*Table 2.* Comparison of object detection and instance segmentation performance on the MS COCO dataset. FLOPs are calculated with input resolution of $1280 \times 800$.

| Backbone | $AP^b$ | $AP^b_{50}$ | $AP^b_{75}$ | $AP^m$ | $AP^m_{50}$ | $AP^m_{75}$ | #Param. | FLOPs |
|---|---|---|---|---|---|---|---|---|
| **Mask R-CNN 1× schedule** | | | | | | | | |
| Swin-T | 42.7 | 65.2 | 46.8 | 39.3 | 62.2 | 42.2 | 48M | 267G |
| DAT-T | 44.4 | 67.6 | 48.5 | 42.4 | 66.1 | 45.5 | 48M | 272G |
| CSWin-T | 46.7 | 68.6 | 51.3 | 42.2 | 65.6 | 45.4 | 42M | 279G |
| ConvNeXt-T | 44.2 | 66.6 | 48.3 | 40.1 | 63.3 | 42.8 | 48M | 262G |
| VMamba-T | 47.3 | 69.3 | 52.0 | 42.7 | 66.4 | 45.9 | 50M | 271G |
| VSSD-T | 47.0 | 69.5 | 51.6 | 42.8 | 66.5 | 45.8 | 52M | 298G |
| MSVMamba-T | 46.9 | 68.7 | 51.4 | 42.5 | 66.2 | 45.8 | 52M | 275G |
| Spatial-Mamba-T | 47.6 | 69.6 | 52.3 | 42.9 | 66.5 | 46.2 | 46M | 261G |
| DefMamba-S | 47.5 | 69.6 | 51.7 | 42.8 | 66.3 | 46.2 | – | 268G |
| MambaOut-T | 45.1 | 67.3 | 49.6 | 41.0 | 64.1 | 44.1 | 43M | 262G |
| Deformba-T | **48.1** | **69.8** | **52.6** | **43.3** | **66.8** | **46.5** | 45M | 266G |
| Swin-S | 44.8 | 66.6 | 48.9 | 40.9 | 63.2 | 44.2 | 69M | 354G |
| DAT-S | 47.1 | 69.9 | 51.5 | 42.5 | 66.7 | 45.4 | 69M | 378G |
| CSWin-S | 47.9 | 70.1 | 52.6 | 43.2 | 67.1 | 46.2 | 54M | 342G |
| ConvNeXt-S | 45.4 | 67.9 | 50.0 | 41.8 | 65.2 | 45.1 | 70M | 348G |
| VMamba-S | 48.7 | 70.0 | 53.4 | 43.7 | 67.3 | 47.0 | 70M | 349G |
| VSSD-S | 48.9 | 71.1 | 53.8 | 43.9 | 68.1 | 47.4 | 77M | 381G |
| MSVMamba-S | 48.1 | 70.1 | 52.8 | 43.2 | 67.3 | 46.5 | 70M | 349G |
| Spatial-Mamba-S | 49.2 | 70.8 | 54.2 | 44.0 | 67.9 | 47.2 | 63M | 315G |
| MambaOut-S | 47.4 | 69.1 | 52.4 | 42.7 | 66.1 | 46.2 | 65M | 354G |
| Deformba-S | **50.0** | **71.5** | **55.0** | **44.7** | **68.7** | **48.3** | 64M | 374G |
| Swin-B | 46.9 | 69.2 | 51.6 | 42.3 | 66.0 | 45.5 | 107M | 496G |
| ViT-Adapter-B | 47.0 | 68.2 | 51.4 | 41.8 | 65.1 | 44.9 | 102M | 557G |
| CSWin-B | 48.7 | 70.4 | 53.9 | 43.9 | 67.8 | 47.3 | 88M | 496G |
| ConvNeXt-B | 47.0 | 69.4 | 51.7 | 42.7 | 66.3 | 46.0 | 111M | 507G |
| VMamba-B | 49.2 | 70.9 | 53.9 | 43.9 | 67.7 | 47.6 | 108M | 485G |
| VSSD-B | 49.9 | 72.1 | 54.7 | 44.8 | 69.1 | 48.3 | 108M | 506G |
| Spatial-Mamba-B | 50.4 | 71.8 | 55.3 | 45.1 | 69.1 | 49.1 | 115M | 494G |
| Deformba-B | **50.6** | **72.0** | **55.8** | **45.2** | **69.3** | **48.9** | 106M | 496G |
| **Mask R-CNN 3× MS schedule** | | | | | | | | |
| Swin-T | 46.0 | 68.1 | 50.3 | 41.6 | 65.1 | 44.9 | 48M | 267G |
| PVTv2-B2 | 47.8 | 69.7 | 52.6 | 43.1 | 66.8 | 46.7 | 45M | 309G |
| ConvNeXt-T | 46.2 | 67.9 | 50.8 | 41.7 | 65.0 | 44.9 | 48M | 262G |
| MSVMamba-T | 48.7 | 69.8 | 53.3 | 43.4 | 67.2 | 46.8 | 52M | 275G |
| VMamba-T | 48.8 | 70.4 | 53.5 | 43.7 | 67.4 | 47.0 | 50M | 271G |
| VSSD-T | 49.1 | 71.2 | 53.5 | 44.0 | 67.6 | 47.4 | 52M | 298G |
| LocalVMamba-T | 48.7 | 70.1 | 53.0 | 43.4 | 67.0 | 46.4 | 45M | 291G |
| Spatial-Mamba-T | 49.3 | 70.7 | 54.3 | 43.8 | 67.8 | 47.2 | 46M | 261G |
| Deformba-T | **50.1** | **71.2** | **54.9** | **44.5** | **68.5** | **48.2** | 45M | 266G |
| Swin-S | 48.2 | 69.8 | 52.8 | 43.2 | 67.0 | 46.1 | 69M | 354G |
| PVTv2-B3 | 48.4 | 69.8 | 53.3 | 43.2 | 66.9 | 46.7 | 65M | 397G |
| ConvNeXt-S | 47.9 | 70.0 | 52.7 | 42.9 | 66.9 | 46.2 | 70M | 348G |
| MSVMamba-S | 49.7 | 70.9 | 54.3 | 44.2 | 68.0 | 47.9 | 70M | 349G |
| VMamba-S | 49.9 | 70.9 | 54.7 | 44.2 | 68.2 | 47.7 | 70M | 349G |
| VSSD-S | 50.7 | 72.2 | 55.8 | 45.1 | 69.4 | 48.8 | 77M | 381G |
| LocalVMamba-S | 49.9 | 70.5 | 54.4 | 44.1 | 67.8 | 47.4 | 69M | 414G |
| Spatial-Mamba-S | 50.5 | 71.5 | 55.7 | 44.6 | 68.7 | 68.7 | 63M | 315G |
| Deformba-S | **50.7** | **71.9** | **55.7** | **45.1** | **69.2** | **48.9** | 64M | 374G |
| Swin-B | 48.6 | 70.0 | 53.4 | 43.3 | 67.1 | 46.7 | 107M | 496G |
| PVTv2-B5 | 48.4 | 69.2 | 52.9 | 42.9 | 66.6 | 46.2 | 102M | 557G |
| ConvNeXt-B | 48.5 | 70.1 | 53.3 | 43.5 | 67.1 | 46.7 | 108M | 486G |
| Deformba-B | **51.8** | **72.4** | **56.8** | **45.6** | **69.8** | **49.3** | 106M | 496G |

83.8%, respectively.

## 5.2. Object Detection and Instance Segmentation

**Settings.** Our evaluation of Deformba utilizes the MS COCO dataset (Lin et al., 2014) within the Mask R-CNN framework (He et al., 2017) for tasks related to object detection and instance segmentation. All experiments are conducted using the MMDetection (Chen et al., 2019) framework. We apply Deformba-T/S/B pretrained on ImageNet-1K as backbones. Following prior work (Liu et al., 2021), we train the model for 12 epochs (1× schedule) and 36

*Table 3.* Comparison of semantic segmentation on the ADE20K dataset. FLOPs are calculated with cropped input resolution of $512 \times 2048$. 'SS' and 'MS' represent single-scale and multi-scale testing, respectively.

| Method | mIoU (SS) | mIoU (MS) | #Params | FLOPs |
|---|---|---|---|---|
| UniRepLKNet-S | 50.5 | 51.0 | 86M | 1036G |
| Swin-S | 47.6 | 49.5 | 81M | 1039G |
| Agent-Swin-S | 48.1 | – | 81M | 1043G |
| ConvNeXt-S | 48.7 | 49.6 | 82M | 1027G |
| NAT-S | 48.0 | 49.5 | 82M | 1010G |
| QFormer$_h$-S | 48.9 | 50.3 | 82M | – |
| PartialFormer-B3 | 48.3 | – | 95M | 1005G |
| MambaOut-S | 49.5 | 50.6 | 76M | 1032G |
| VMamba-S | **50.6** | 51.2 | 82M | 1028G |
| LocalVMamba-S | 50.0 | 51.0 | 81M | 1095G |
| Deformba-S | **50.6** | **51.7** | 75M | 1052G |
| Swin-B | 48.1 | 49.7 | 121M | 1188G |
| Agent-Swin-B | 48.7 | – | 121M | 1196G |
| ConvNeXt-B | 49.1 | 49.9 | 122M | 1170G |
| NAT-B | 48.5 | 49.7 | 123M | 1137G |
| QFormer$_h$-B | 49.5 | 50.6 | 123M | – |
| MambaOut-B | 49.6 | 51.0 | 112M | 1178G |
| VMamba-B | 51.0 | 51.6 | 122M | 1170G |
| Deformba-B | **52.0** | **52.6** | 117M | 1177G |

epochs (3× schedule) with multi-scale inputs. Optimization is to use the AdamW optimizer, with an initial learning rate of 0.0001 and a batch size of 16 and also MESA (Du et al., 2022).

**Results.** Table 2 details the results on the MS COCO dataset. All variants of Deformba demonstrate comparable or superior performance in various configurations. For 3× schedule, Deformba-B demonstrates a significant advantage whcih achieves a bounding box mAP of 51.8% and a mask mAP of 69.8%, outperforming ConvNeXt-B (Liu et al., 2022), PVTv2-B5 (Wang et al., 2022a) and Swin-B (Liu et al., 2021) by 3.3%, 3.4% and 3.2% in bounding box mAP and 2.7%, 3.2% and 2.7% in mask mAP with comparable or fewer parameters and FLOPs, respectively. Furthermore, our method demonstrates superior performance to other methods under the same configuration. The 1× multi-scale training schedule has the same trends.

## 5.3. Semantic Segmentation

**Settings.** We evaluate the sematic segmentation performance using the ADE20K dataset (Zhou et al., 2017). Our experiments adopt the widely used UPerNet segmentor (Xiao et al., 2018) and the MMSegmentation framework (Contributors, 2020). The backbone is pretrained on ImageNet-1K followed by previous work (Liu et al., 2021). This training process includes 160K iterations with a batch size of 16 and model optimization uses the AdamW optimizer with a weight decay of 0.05 and a learning rate of $6 \times 10^{-5}$. All experiments were conducted using a default input resolution of $512 \times 512$.

| Method | Backbone | # Frames | NDS↑ | mAP↑ | mATE↓ | mASE↓ | mAOE↓ | mAVE↓ | mAAE↓ |
|---|---|---|---|---|---|---|---|---|---|
| BEVFormerV1-T | ResNet50 | 3 | 0.354 | 0.252 | 0.900 | 0.294 | 0.655 | 0.657 | 0.216 |
| BEVFormerV2-T* | ResNet50 | 3 | 0.397 | 0.270 | 0.820 | 0.301 | 0.594 | 0.469 | 0.195 |
| BEVDiffuser | ResNet50 | 3 | 0.391 | 0.283 | 0.859 | 0.285 | 0.558 | 0.592 | 0.212 |
| MamBEV-T | ResNet50 | 3 | 0.399 | 0.266 | 0.794 | 0.298 | 0.575 | 0.469 | 0.199 |
| Deformba-T | ResNet50 | 3 | **0.405** | 0.276 | 0.823 | 0.296 | 0.564 | 0.460 | 0.189 |
| FCOS3D | ResNet101 | 1 | 0.415 | 0.343 | 0.725 | 0.263 | 0.422 | 1.292 | 0.153 |
| Focal-PETR | ResNet101 | 1 | 0.461 | 0.390 | 0.678 | 0.263 | 0.395 | 0.804 | 0.202 |
| DETR3D | ResNet101 | 1 | 0.434 | 0.349 | 0.716 | 0.268 | 0.379 | 0.842 | 0.200 |
| PolarDETR | ResNet101 | 2 | 0.488 | 0.383 | 0.707 | 0.269 | 0.344 | 0.518 | 0.196 |
| PolarFormer | ResNet101 | 2 | 0.528 | 0.432 | 0.648 | 0.270 | 0.348 | 0.409 | 0.201 |
| BEVFormerV1-S | ResNet101 | 3 | 0.479 | 0.370 | 0.721 | 0.279 | 0.407 | 0.436 | 0.220 |
| BEVFormerV1-B | ResNet101 | 4 | 0.517 | 0.416 | 0.673 | 0.274 | 0.372 | 0.394 | 0.198 |
| MamBEV-S | ResNet101 | 4 | 0.525 | 0.423 | 0.662 | 0.280 | 0.386 | 0.354 | 0.183 |
| BEVFormer-B-Enh | ResNet101 | 4 | 0.530 | 0.419 | 0.632 | 0.268 | 0.328 | 0.374 | 0.195 |
| Deformba-S | ResNet101 | 3 | **0.525** | 0.432 | 0.650 | 0.280 | 0.369 | 0.410 | 0.203 |
| Deformba-S | ResNet101 | 4 | **0.538** | 0.445 | 0.642 | 0.273 | 0.352 | 0.382 | 0.202 |

*Table 4.* Comparison of BEV 3D detection performance on the nuScenes val dataset. * indicates models trained by us.

**Results.** The results on semantic segmentation are shown in Table 3. Our Deformba-S and Deformba-B demonstrate impressive performance. Deformba-B achieves a mIoU of 52.0 and 52.6 in single-scale and multi-scale, respectively, which outperforms VMamba-B, ConvNeXt-B, MambaOut-B. Deformba-S achieves comparable performance to VMamba-S in single-scale testing, but outperfoms in multi-scale testing.

### 5.4. 3D BEV Detection

**Settings.** We follow the methodologies of the previous work of (Wang et al., 2022b; Li et al., 2022a) and (Yang et al., 2023). We evaluate on two backbones: ResNet101 with deformable convolution layers and ResNet50 which are pretrained on a depth prediction task and COCO, respectively. During training, the first stage of the backbone is frozen, and all other stages are trained at a 10% learning rate to fine-tune their latent representations to the multiview autonomous driving setting. We evaluate on the nuScenes dataset (Caesar et al., 2020), a large-scale multi-modal autonomous driving benchmark comprising 1,000 urban driving scenes recorded in Boston and Singapore. In our method and experiments, only the camera frames, sensor calibration data, and GPS data from nuScenes dataset are used in making predictions.

**Results.** We present a comparison of our results in Table 4 against state-of-the-art methods at compatible parameter and image input scales. We only use camera features; additionally, we do not make use of any auxiliary loss as in the works of Yang et al. (2023). We aligned the definitions of tiny and small models with BEVFormerV1. The CNN based temporal component of our model consists of 57M parameters for the small configuration and 32M parameters for the tiny configuration. Excluding the temporal

| Cross Attention Module | BEV Scale | Feature Map Size | Params (K) | GFLOPs |
|---|---|---|---|---|
| | $50 \times 50$ | $6 \times 25 \times 15$ | 176 | 2.1 |
| Deformba | $100 \times 100$ | $6 \times 40 \times 24$ | 176 | 7.6 |
| | $200 \times 200$ | $6 \times 50 \times 29$ | 176 | 26 |
| | $50 \times 50$ | $6 \times 25 \times 15$ | 239 | 3.7 |
| XQSSM | $100 \times 100$ | $6 \times 40 \times 24$ | 239 | 14 |
| | $200 \times 200$ | $6 \times 50 \times 29$ | 239 | 51 |
| | $50 \times 50$ | $6 \times 25 \times 15$ | 156 | 3.3 |
| Deformable | $100 \times 100$ | $6 \times 40 \times 24$ | 156 | 12.8 |
| | $200 \times 200$ | $6 \times 50 \times 29$ | 156 | 49.5 |
| | $50 \times 50$ | $6 \times 25 \times 15$ | 263 | 23.9 |
| Dot Product | $100 \times 100$ | $6 \times 40 \times 24$ | 263 | 228.8 |
| | $200 \times 200$ | $6 \times 50 \times 29$ | 263 | 1,432.5 |

*Table 5.* Scaling of cross attention modules, where each evaluated under three BEV and image resolutions.

| Module design | Params (M) | FLOPs (G) | Top-1(%)↑ |
|---|---|---|---|
| Baseline | 25.0M | 4.7G | 80.8 |
| + CASF w/o ECA | 25.5M | 4.8G | 81.3 |
| + CASF w/ ECA | 25.5M | 4.8G | 81.4 |
| + MESA | 25.5M | 4.8G | 81.6 |

*Table 6.* Ablation studies on Deformba-T for module designs

portion, our Deformba-S model has 62M parameters, while Deformba-T has 38M parameters. For comparison, BEVFormer's tiny, small, and base models have 34, 60, and 69 million parameters respectively. Deformba-T outperforms transformer-based BEVFormerV1-T and BEVFormerV2-T by 0.051 (+12.6%) and 0.008 (+2.0%) under similar conditions for the ResNet50 backbone. Moreover, Deformba-T has performance comparable to the recently introduced SSM-based MamBEV-T. We report an increase of 0.059 (11.0%) and 0.013 (2.4%), Deformba-Small with 4 frames, in NDS over the previous BEVFormerV1-S model and MamBEV-S model and 0.021 (3.9%) increase over the previous BEVFormerV1-B model. This demonstrates the effectiveness of our methods.

In Table 5, we compute estimated FLOPs and parameters for

| Method | Param. (M) | FLOPs (G) | Top-1(%)↑ |
|---|---|---|---|
| Zero-offset | 25.0 | 4.7 | 80.8 |
| Fixed offset | 25.0 | 4.8 | 80.9 |
| MLP | 30.8 | 5.6 | 81.0 |
| Dynamic offset | 25.5 | 4.8 | 81.3 |

*Table 7.* Ablation studies of the deformable sampling mechanism. Dynamic offset corresponds to our learned deformable sampling design.

| Conv Type | Scan | mAP | $mAP_{50}$ | $mAP_{75}$ |
|---|---|---|---|---|
| Non-causal | Single | 0.393 | 0.575 | 0.431 |
| | Bidirectional | 0.393 | 0.576 | 0.430 |
| Causal | Single | 0.388 | 0.570 | 0.426 |
| | Bidirectional | 0.390 | 0.571 | 0.425 |
| No Conv | Single | 0.386 | 0.565 | 0.422 |
| | Bidirectional | 0.390 | 0.575 | 0.427 |
| No State | Single | 0.384 | 0.569 | 0.425 |

*Table 8.* Convolution Configurations

model configurations which use our Deformba, XQSSM (Ke et al., 2025b), deformable attention, or standard dot product attention. Deformba, XQSSM and deformable attention scale linearly in computational complexity with respect to the size of the inputs V and Q, though the coefficient factor of Deformba is smaller.

### 5.5. Ablation Studies

To evaluate the effectiveness of our method, we conduct ablations of key components of Deformba. As shown in Table 6, we conducted an ImageNet-1K ablation study under a 100 epoch training schedule. The baseline adopts a single sweep scanning strategy and does not include CASF. Employing CASF w/o ECA and with ECA, the top-1 accuracy is improved by 0.5% and 0.6%, respectively, while having negligible cost. Furthermore, by using MESA to mitigate overfitting, the top-1 accuracy is increased by 0.2%. In Table 8, we study the design choices in Mamba implementation in the COCO dataset, including the type of convolution layer, scanning directions, and the role of the hidden state. The results indicate that using a non-causal convolution improves accuracy, bidirectional scanning is not necessary for accuracy gains, and the hidden state is important.

To fully isolate which part of CASF is responsible for the gain, we add more targeted controls, as shown in Table 7. We remove ECA and MESA, keep the write-read framework unchanged, and replace the learned deformable sampling with predefined non-adaptive alternatives or an MLP-based fusion module. The compared variants are: (i) *zero-offset sampling*, where each location only reads from its own reference point; (ii) *fixed-offset sampling*, where learned offsets are replaced by a predefined 1-hop local stencil around each reference point $\mathcal{E}(i, j)$; and (iii) *MLP fusion*, where the sampling mechanism is replaced by an MLP such as $\hat{S}_Q = W_2\text{GLUE}\left(W_1\text{LN}(S_{2D})\right)$. The results show that the improvement comes directly from adaptive deformable state reading, rather than from increased capacity or from the existence of an extra fusion module alone.

Moreover, our argument is that the limitation of prior vision SSMs is not only using a fixed scan, but letting the scan path itself determine the final interaction topology. In our Deformba, the single scan is used only for a compact write pass that constructs a shared state memory $\mathbf{S}_{2D}$, which flatten

| Method | Top-1 acc. (%) |
|---|---|
| Unidi-Scan | 81.33 |
| Bidi-Scan | 81.24 |
| Cascade-Scan Row and Col | 81.28 |
| Cross Scan | 81.38 |

*Table 9.* Scan method ablation

2D images to 1D sequences so that they can be processed by SSM. The final spatial interaction is then established by CASF through adaptive reading from the written state map. Therefore CASF reduces the dependence of spatial modeling on scan direction by augmenting the fixed write graph with learned read connections. This is consistent with our original ablation in Table 8, where single and bidirectional traversals achieve the same mAP (0.393). To further verify this point, we additionally evaluate different scan patterns on ImageNet-1K using Deformba-T (following the scan settings of VMamba). As shown in Table 9, the results are highly similar across different scan patterns. This suggests once CASF is introduced, the model becomes insensitive to specific scanning strategy, and the spatial interaction is governed mainly by adaptive reading rather than by the predefined scan order.

More ablations are provided in the Appendix section A.

## 6. Conclusion

In this work, we propose Deformba, a vision state space model architecture that augments Mamba with Context-Adaptive State Fusion (CASF). By decoupling the SSM write operation from the read operation, CASF exposes an intermediate hidden state representation that supports both self-attention-style mixing and query-to-source fusion, analogous to cross-attention, without sacrificing the efficiency of SSMs. We evaluate Deformba across a broad range of 2D and 3D tasks. Extensive experiments and ablation studies demonstrate our method is robust across design variations while maintaining both effectiveness and efficiency.

## Acknowledgment

Research was sponsored by the Army Research Laboratory and was accomplished under Cooperative Agreement Number W911NF-23-2-0224 and supported in part by funds from Toyota Motor North America. The views and conclusions contained in this document are those of the authors and should not be interpreted as representing the official policies, either expressed or implied, of the Army Research Laboratory or the U.S. Government. The U.S. Government is authorized to reproduce and distribute reprints for Government purposes notwithstanding any copyright notation herein. The contents do not necessarily reflect the official views of Toyota Motor North America.

## Impact Statement

This paper presents work whose goal is to advance the field of machine learning. There are many potential societal consequences of our work, none of which we feel must be specifically highlighted here.

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

Contributors, M. Mmsegmentation: Openmmlab semantic segmentation toolbox and benchmark, 2020.

Dao, T. and Gu, A. Transformers are ssms: Generalized models and efficient algorithms through structured state space duality. *arXiv preprint arXiv:2405.21060*, 2024.

Deng, J., Dong, W., Socher, R., Li, L.-J., Li, K., and Fei-Fei, L. Imagenet: A large-scale hierarchical image database. In *2009 IEEE conference on computer vision and pattern recognition*, pp. 248–255. Ieee, 2009.

Du, J., Zhou, D., Feng, J., Tan, V., and Zhou, J. T. Sharpness-aware training for free. *Advances in Neural Information Processing Systems*, 35:23439–23451, 2022.

Feng, Y., Hu, J., Lu, Q., Niu, J., Tan, L., Yuan, S., Yan, Z., Jia, Y., He, Q., Ge, S., et al. Muvr: A multi-modal untrimmed video retrieval benchmark with multi-level visual correspondence. *Advances in Neural Information Processing Systems*, 38, 2026.

Gu, A. and Dao, T. Mamba: Linear-time sequence modeling with selective state spaces. *arXiv preprint arXiv:2312.00752*, 2023.

Gu, A., Dao, T., Ermon, S., Rudra, A., and Ré, C. Hippo: Recurrent memory with optimal polynomial projections. *Advances in neural information processing systems*, 33: 1474–1487, 2020.

Gu, A., Goel, K., and Ré, C. Efficiently modeling long sequences with structured state spaces. *arXiv preprint arXiv:2111.00396*, 2021a.

Gu, A., Johnson, I., Goel, K., Saab, K., Dao, T., Rudra, A., and Ré, C. Combining recurrent, convolutional, and continuous-time models with linear state space layers. *Advances in neural information processing systems*, 34: 572–585, 2021b.

He, K., Gkioxari, G., Dollár, P., and Girshick, R. Mask r-cnn. In *Proceedings of the IEEE international conference on computer vision*, pp. 2961–2969, 2017.

Huang, S., Lu, Z., Cun, X., Yu, Y., Zhou, X., and Shen, X. Deim: Detr with improved matching for fast convergence. In *Proceedings of the Computer Vision and Pattern Recognition Conference*, pp. 15162–15171, 2025.

Huang, T., Pei, X., You, S., Wang, F., Qian, C., and Xu, C. Localmamba: Visual state space model with windowed selective scan. In *European Conference on Computer Vision*, pp. 12–22. Springer, 2024.

Jiang, C., Zhou, D., Liu, J., and Sun, K. L. Vectorworld: Efficient streaming world model via diffusion flow on vector graphs. *arXiv preprint arXiv:2603.17652*, 2026.

Ke, H., Morris, J., Liu, Y., Kitai, S., Oguchi, K., Ding, Y., and Wang, H. Tinybev: Compact temporal fusion for multi-view 3d perception. In *Proceedings of the Tenth ACM/IEEE Symposium on Edge Computing*, pp. 1–6, 2025a.

Ke, H., Morris, J., Oguchi, K., Cao, X., Liu, Y., Wang, H., and Ding, Y. Mambev: Enabling state space models to learn birds-eye-view representations. In *The Thirteenth International Conference on Learning Representations*, 2025b.

Kuhn, H. W. The hungarian method for the assignment problem. *Naval research logistics quarterly*, 2(1-2):83–97, 1955.

Lee, S., Choi, J., and Kim, H. J. Efficientvim: Efficient vision mamba with hidden state mixer based state space duality. In *Proceedings of the Computer Vision and Pattern Recognition Conference*, pp. 14923–14933, 2025.

Li, T., Li, C., Lyu, J., Pei, H., Zhang, B., Jin, T., and Ji, R. Damamba: Vision state space model with dynamic adaptive scan. *arXiv preprint arXiv:2502.12627*, 2025.

Li, Z., Wang, W., Li, H., Xie, E., Sima, C., Lu, T., Qiao, Y., and Dai, J. Bevformer: Learning bird's-eye-view representation from multi-camera images via spatiotemporal transformers. *arXiv preprint arXiv:2203.17270*, 2022a.

Li, Z., Wang, W., Xie, E., Yu, Z., Anandkumar, A., Alvarez, J. M., Luo, P., and Lu, T. Panoptic segformer: Delving deeper into panoptic segmentation with transformers. In *Proceedings of the IEEE/CVF conference on computer vision and pattern recognition*, pp. 1280–1289, 2022b.

Liang, G., Wang, Z., Hu, J., Zhou, H., Xue, Z., Zhang, J., Xu, D., and Yu, Q. Render-in-the-loop: Vector graphics generation via visual self-feedback. *arXiv preprint arXiv:2604.20730*, 2026a.

Liang, G., Wang, Z., Wang, C., Hu, J., Zhou, H., Liu, J., Zhang, J., Xu, D., and Yu, Q. Vanim: Rendering-aware sparse state modeling for structure-preserving vector animation. *arXiv preprint arXiv:2605.01517*, 2026b.

Lin, H., Liu, Z., Zhu, Y., Qin, C., Lin, J., Shang, X., He, C., Zhang, W., and Wu, L. Mmfinereason: Closing the multimodal reasoning gap via open data-centric methods. *arXiv preprint arXiv:2601.21821*, 2026.

Lin, T.-Y., Maire, M., Belongie, S., Hays, J., Perona, P., Ramanan, D., Dollár, P., and Zitnick, C. L. Microsoft coco: Common objects in context. In *European conference on computer vision*, pp. 740–755. Springer, 2014.

Lin, T.-Y., Dollár, P., Girshick, R., He, K., Hariharan, B., and Belongie, S. Feature pyramid networks for object detection. In *Proceedings of the IEEE conference on computer vision and pattern recognition*, pp. 2117–2125, 2017a.

Lin, T.-Y., Goyal, P., Girshick, R., He, K., and Dollár, P. Focal loss for dense object detection. In *Proceedings of the IEEE international conference on computer vision*, pp. 2980–2988, 2017b.

Liu, L., Zhang, M., Yin, J., Liu, T., Ji, W., Piao, Y., and Lu, H. Defmamba: Deformable visual state space model. In *Proceedings of the Computer Vision and Pattern Recognition Conference*, pp. 8838–8847, 2025a.

Liu, Y., Yan, J., Jia, F., Li, S., Gao, A., Wang, T., and Zhang, X. Petrv2: A unified framework for 3d perception from multi-camera images. In *Proceedings of the IEEE/CVF International Conference on Computer Vision*, pp. 3262–3272, 2023.

Liu, Y., Tian, Y., Zhao, Y., Yu, H., Xie, L., Wang, Y., Ye, Q., Jiao, J., and Liu, Y. Vmamba: Visual state space model. *Advances in neural information processing systems*, 37: 103031–103063, 2024.

Liu, Z., Lin, Y., Cao, Y., Hu, H., Wei, Y., Zhang, Z., Lin, S., and Guo, B. Swin transformer: Hierarchical vision transformer using shifted windows. In *Proceedings of the IEEE/CVF international conference on computer vision*, pp. 10012–10022, 2021.

Liu, Z., Mao, H., Wu, C.-Y., Feichtenhofer, C., Darrell, T., and Xie, S. A convnet for the 2020s. In *Proceedings of the IEEE/CVF conference on computer vision and pattern recognition*, pp. 11976–11986, 2022.

Liu, Z., Liu, M., Chen, J., Xu, J., Cui, B., He, C., and Zhang, W. Fusion: Fully integration of vision-language representations for deep cross-modal understanding. *arXiv preprint arXiv:2504.09925*, 2025b.

Ma, X., Ni, Z., and Chen, X. Tinyvim: Frequency decoupling for tiny hybrid vision mamba. In *Proceedings of the IEEE/CVF International Conference on Computer Vision*, pp. 23519–23529, 2025.

Schlag, I., Irie, K., and Schmidhuber, J. Linear transformers are secretly fast weight programmers. In *International conference on machine learning*, pp. 9355–9366. PMLR, 2021.

Shaker, A., Wasim, S. T., Khan, S., Gall, J., and Khan, F. S. Groupmamba: Efficient group-based visual state space model. In *Proceedings of the Computer Vision and Pattern Recognition Conference*, pp. 14912–14922, 2025.

Shi, D. Transnext: Robust foveal visual perception for vision transformers. In *Proceedings of the IEEE/CVF conference on computer vision and pattern recognition*, pp. 17773–17783, 2024.

Shi, Y., Dong, M., and Xu, C. Multi-scale vmamba: Hierarchy in hierarchy visual state space model. *Advances in Neural Information Processing Systems*, 37:25687–25708, 2024.

Shi, Y., Li, M., Dong, M., and Xu, C. Vssd: Vision mamba with non-causal state space duality. In *Proceedings of the IEEE/CVF International Conference on Computer Vision*, pp. 10819–10829, 2025.

Touvron, H., Cord, M., Douze, M., Massa, F., Sablayrolles, A., and Jégou, H. Training data-efficient image transformers & distillation through attention. In *International conference on machine learning*, pp. 10347–10357. PMLR, 2021.

Wang, Q., Wu, B., Zhu, P., Li, P., Zuo, W., and Hu, Q. Eca-net: Efficient channel attention for deep convolutional neural networks. In *Proceedings of the IEEE/CVF conference on computer vision and pattern recognition*, pp. 11534–11542, 2020.

Wang, T., Zheng, X., and Cai, Z. Neighborhood pseudo-graph based personalized federated graph learning. *IEEE Transactions on Mobile Computing*, 2026.

Wang, W., Xie, E., Li, X., Fan, D.-P., Song, K., Liang, D., Lu, T., Luo, P., and Shao, L. Pvt v2: Improved baselines with pyramid vision transformer. *Computational visual media*, 8(3):415–424, 2022a.

Wang, Y., Guizilini, V. C., Zhang, T., Wang, Y., Zhao, H., and Solomon, J. Detr3d: 3d object detection from multi-view images via 3d-to-2d queries. In *Conference on robot learning*, pp. 180–191. PMLR, 2022b.

Xiao, C., Li, M., Zhang, Z., Meng, D., and Zhang, L. Spatial-mamba: Effective visual state space models via structure-aware state fusion. *arXiv preprint arXiv:2410.15091*, 2024a.

Xiao, T., Liu, Y., Zhou, B., Jiang, Y., and Sun, J. Unified perceptual parsing for scene understanding. In *Proceedings of the European conference on computer vision (ECCV)*, pp. 418–434, 2018.

Xiao, Y., Song, L., Huang, S., Wang, J., Song, S., Ge, Y., Li, X., and Shan, Y. Grootvl: Tree topology is all you need in state space model. *arXiv preprint arXiv:2406.02395*, 2024b.

Xu, B., Dai, X., Tang, D., and Zhang, K. One surrogate to fool them all: Universal, transferable, and targeted adversarial attacks with clip. In *Proceedings of the 2025 ACM SIGSAC Conference on Computer and Communications Security*, pp. 3087–3101, 2025a.

Xu, B., Yang, F., Dai, X., Tang, D., and Zhang, K. From internal diagnosis to external auditing: A vlm-driven paradigm for online test-time backdoor defense. *arXiv preprint arXiv:2601.19448*, 2026.

Xu, H., Ke, X., Li, Y., Xu, R., Wu, H., Lin, X., and Guo, W. Vision-language action knowledge learning for semantic-aware action quality assessment. In *European conference on computer vision*, pp. 423–440. Springer, 2024.

Xu, H., Ke, X., Wu, H., Xu, R., Li, Y., and Guo, W. Language-guided audio-visual learning for long-term sports assessment. In *Proceedings of the Computer Vision and Pattern Recognition Conference*, pp. 23967–23977, 2025b.

Xu, H., Wu, H., Ke, X., Li, Y., Xu, R., and Guo, W. Quality-guided vision-language learning for long-term action quality assessment. *IEEE Transactions on Multimedia*, 2025c.

Yang, C., Chen, Y., Tian, H., Tao, C., Zhu, X., Zhang, Z., Huang, G., Li, H., Qiao, Y., Lu, L., et al. Bevformer v2: Adapting modern image backbones to bird's-eye-view recognition via perspective supervision. In *Proceedings of the IEEE/CVF Conference on Computer Vision and Pattern Recognition*, pp. 17830–17839, 2023.

Yang, C., Chen, Z., Espinosa, M., Ericsson, L., Wang, Z., Liu, J., and Crowley, E. J. Plainmamba: Improving non-hierarchical mamba in visual recognition. *arXiv preprint arXiv:2403.17695*, 2024a.

Yang, S., Wang, B., Zhang, Y., Shen, Y., and Kim, Y. Parallelizing linear transformers with the delta rule over sequence length. *Advances in neural information processing systems*, 37:115491–115522, 2024b.

Yu, W. and Wang, X. Mambaout: Do we really need mamba for vision? In *Proceedings of the Computer Vision and Pattern Recognition Conference*, pp. 4484–4496, 2025.

Zhang, G., Fan, L., He, C., Lei, Z., ZHANG, Z.-X., and Zhang, L. Voxel mamba: Group-free state space models for point cloud based 3d object detection. *Advances in Neural Information Processing Systems*, 37:81489–81509, 2024.

Zhou, B., Zhao, H., Puig, X., Fidler, S., Barriuso, A., and Torralba, A. Scene parsing through ade20k dataset. In *Proceedings of the IEEE conference on computer vision and pattern recognition*, pp. 633–641, 2017.

Zhu, C., Lin, Y., Chen, S., Wang, Y., and Lin, J. Medeyes: Learning dynamic visual focus for medical progressive diagnosis. In *Proceedings of the AAAI Conference on Artificial Intelligence*, volume 40, pp. 13916–13924, 2026a.

Zhu, C., Zeng, J., Jiang, J., Lin, J., and Wang, Y. Medsynapse-v: Bridging visual perception and clinical intuition via latent memory evolution. *arXiv preprint arXiv:2604.26283*, 2026b.

Zhu, L., Liao, B., Zhang, Q., Wang, X., Liu, W., and Wang, X. Vision mamba: Efficient visual representation learning with bidirectional state space model. *arXiv preprint arXiv:2401.09417*, 2024.

Zhu, X., Su, W., Lu, L., Li, B., Wang, X., and Dai, J. Deformable detr: Deformable transformers for end-to-end object detection. *arXiv preprint arXiv:2010.04159*, 2020.

# A. Appendix

## A.1. Deformba Cross Attention Method

The general problem setup is we aim to take a set of input image feature maps $F$ and a set of queries $B$. The module is tasked with learning a function $f$ which selects information from $F$ to update $B$ in a supervised learning problem. This process is done in 3 main steps 1) construction of hidden states of $F$, 2) construction of hidden states of $B$, and 3) decoding of hidden states of $B$. In the case of BEV, the output is an updated representation of the BEV grid and for COCO the output is an updated set of object queries.

**BEV Method Overview.** A visualization of our overall method for learning a BEV construction is provided in Figure 5. Many of the existing components follow the methodologies of (Li et al., 2022a), (Liu et al., 2023), and (Ke et al., 2025b). The encoding function $f_{enc}$ is parameterized by the ResNet backbone, feature pyramid network (FPN), BEV encoder, and temporal fusion modules. The pipeline has three main stages as follows. First, image feature maps of different scales $F_0, ..., F_i$ are produced as intermediate backbone outputs. Second, their channels are reduced to a uniform size $D$ through horizontal convolutions in the FPN (Lin et al., 2017a). Then, feature representations and interactions are modeled using a novel SSM-based pipeline (discussed below). Lastly, the decoding function $f_{dec}$ operates on this representation to make predictions. As in (Zhu et al., 2020) (Li et al., 2022a) (Yang et al., 2023) methods, the decoder uses alternating layers of grouped object query self-attention and object query to BEV feature deformable cross-attention. Decoder heads at each layer follow the masked decoder heads from (Li et al., 2022b) which predict a bounding box, its properties, and the object class. We do not use a map segmentation head, and only use the detection head which is optimized using 3D Hungarian (Kuhn, 1955) set matching with smooth $L_1$ loss for the bounding boxes and focal loss (Lin et al., 2017b) for class prediction. We replace the cross attention module in encoder, we next discuss the details.

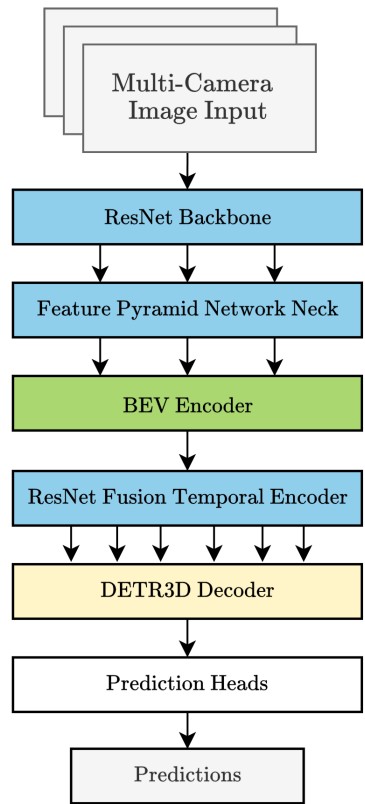

*Figure 5.* Illustration of BEV architecture.

### A.1.1. DEFORMBA CROSS ATTENTION

We decouple the SSM write operation from the SSM read operation, applying them to the image features and query features, respectively. Figure 4 an illustration of this method is displayed for the multi-view case.

### A.1.2. STATE MAP GENERATION

We first leverage the SSM's write recurrence from SSM wirte equation to encode the entire 2D image feature map $F_0, ..., F_i$. This operation generates a dense, context-aware hidden state map, $S_{F_i}$. Each pixel in this state map $S_{F_i}$ is no longer a local feature but a hidden state that summarizes the information scanned up to that point.

### A.1.3. DEFORMABLE STATE SAMPLING

This stage performs a deformable sampling operation, where the queries $v_B$ attend to the state map $S_{F_i}$. To learn the depth information, following BEVFormer, we lift each 2D BEV location $(x, y)$ into a 3D pillar $(x, y, z)$ and project $Z$ evenly spaced pillar points onto each image. The projected 2D locations are used as reference points, which indicate where an object at the corresponding BEV location would appear in each camera view according to the ego vehicle's camera calibration. We denote the set of reference points that fall inside the image bounds and have positive depth as $\mathcal{E}_{hit}$.

The BEV query $v_B$ is used to generate a set of offsets $\Delta p$ through a linear layer. For each valid reference point in $\mathcal{E}_{hit}$, we

add the learned offsets to obtain the deformable sampling locations $\mathcal{P}$. Since these sampling locations are continuous, we use bilinear interpolation to extract the corresponding hidden states from the image state map $S_{F_i}$. The sampled visual evidence is then fused back into the BEV query state with weights $w$ learned by a linear layer from $S_B$. The update process is formulated as:

$$\mathcal{P} = \{e + \Delta p_k \mid e \in \mathcal{E}_{hit}, \ k = 1, \ldots, K\}, \tag{8}$$

$$S_B = \frac{1}{|\mathcal{P}|} \sum_{p_n \in \mathcal{P}} w_n \cdot \phi(S_{F_i}, p_n), \tag{9}$$

where $\phi(S_{F_i}, p_n)$ denotes bilinear interpolation at the continuous sampling location $p_n$ on the image hidden state map $S_{F_i}$, and $K$ is the number of learned offset points around each valid reference point. Therefore, the number of sampling locations attended in a single-view map is $|\mathcal{P}| = |\mathcal{E}_{hit}| \times K$.

### A.1.4. SSM DECODE

The final output is generated from the enriched query state $S_B$ using the output projection equation $O = S_B Q + V D$, where Q and D are the projection parameters associated with V.

### A.2. BEV Implementation Details

For experiments on nuScenes dataset, we used a learning rate of $8 \times 10^{-4}$, with a linear warmup for 10% of the scheduled steps starting from $\frac{8}{3} \times 10^{-4}$ Following the warmup, the learning rate follows an epoch based cosine annealing schedule with a minimum learning rate of $8 \times 10^{-7}$. We trained with an effective batch size of 32 with no gradient accumulation on 8 A6000s for 30 epochs, truncated at 24 epochs. An exponential moving average according to the function $w'_t = (1 - 0.0002)w_t + 0.0002w_t$ is applied to all weights starting from the beignning of training. An AdamW optimizer with a 0.01 weight decay is used, and training employs an automatic mixed precision optimizer wrapper with an initial gradient scaling of 512. A 0.1 multiplier is applied to the learning rate of the backbone weights and the deformable attention sampling offsets (Zhu et al., 2020). We train the models from scratch using a randomly initialized network for the encoder layers. For experiments on the COCO dataset, we trained each decoder configuration using 3 decoder layers for 5 epochs at a learning rate of $2 \times 10^{-4}$ and 5 epochs at $2 \times 10^{-5}$. For the main COCO results, we trained each model for 10 epochs at a learning rate of $1 \times 10^{-4}$ and 5 epochs at $1 \times 10^{-5}$. The resulting models were evaluated on the COCO evaluation dataset to obtain the mean average precision across different IoU thresholds and bounding box sizes.

### A.3. Hyperparameter Tuning

We perform a univariate analysis of key hyperparameters to optimize the Deformba block.

**Feedforward Channels:** We observe that increasing the channel dimension of the Feedforward Network (FFN) from 256 to 1024 improves performance from 0.397 to 0.401 mAP (Table 10). This suggests that the Deformba block benefits from increased capacity to process the aggregated state features.

*Table 10.* FFN Channels Ablation

| FFN Channels | mAP | mAP$_{50}$ | mAP$_{75}$ | Params (M) |
|---|---|---|---|---|
| None (Default) | 0.393 | 0.575 | 0.431 | **2.02** |
| 256 | 0.397 | 0.578 | 0.436 | 2.41 |
| 512 | 0.395 | 0.577 | 0.433 | 2.81 |
| 1024 | **0.401** | **0.583** | **0.442** | 3.59 |

**Expansion Factor:** We explored the impact of the expansion factor within the Deformba layer (Table 11). Increasing the expansion factor from 1.0 to 2.0 improves accuracy. As noted in our implementation details, increasing the expand factor while reducing or removing the linear projection size is a strategy we employ to maximize returns on parameter count.

**Attention Heads:** Unlike standard Transformers, increasing the number of heads in the sampling portion of the Deformba block does not yield significant improvements, with 2, 4, and 8 heads performing comparably (Table 12). We utilize 8 heads in the default configuration to align with the baseline.

*Table 11.* Expand Factor Ablation

| Expand Factor | mAP | mAP$_{50}$ | mAP$_{75}$ | Params (M) |
|---|---|---|---|---|
| 0.5 | 0.380 | 0.561 | 0.415 | **1.56** |
| 1.0 (Default) | 0.393 | 0.575 | 0.431 | 2.02 |
| 2.0 | **0.398** | **0.578** | **0.433** | 2.94 |

*Table 12.* Attention Heads Ablation

| Heads | mAP | mAP$_{50}$ | mAP$_{75}$ | Params (M) |
|---|---|---|---|---|
| 2 | **0.395** | 0.577 | **0.435** | **1.80** |
| 4 | 0.394 | **0.578** | 0.434 | 1.87 |
| 8 (Default) | 0.393 | 0.575 | 0.431 | 2.02 |

**BEV Training Setting:** In Table 13, we provide the hyper-parameters and training recipes of Deformba used for ResNet-50 and Resnet-101 backbones.

*Table 13.* BEV Training settings of Deformba with ResNet backbones for the main results.

| backbone | ResNet-50 | ResNet-101 |
|---|---|---|
| image size | $480 \times 800$ | $768 \times 1280$ |
| FFN | ✗ | ✗ |
| training epochs | 30 | 30 |
| effective batch size | 32 | 32 |
| optimizer | AdamW | AdamW |
| base learning rate | 8e-4 | 8e-4 |
| weight decay | 0.01 | 0.01 |
| lr schedule | CosineAnnealingLR | CosineAnnealingLR |
| warmup epochs | 3 | 3 |
| warmup schedule | linear | linear |
| bbox loss weight | 2.0 | 2.0 |
| bbox loss weight | 0.8 | 0.8 |
| gradient clip | 35 | 35 |
| expand | 1 | 1 |
| sampling head | 8 | 8 |
| conv1d | non-causal | non-causal |
| dt rank | 16 | 32 |
| traversal | 1 | 1 |

# B. Efficiency

### B.0.1. IO

Our approach leverages the hardware-aware parallel scan to mitigate excessive IO; however, a key distinction in our architecture is the necessity of state materialization. Unlike standard Mamba implementations that may fuse the scan and output projection to bypass writing intermediate states to HBM, the Deformba block explicitly materializes the hidden states $\mathbf{S}_{2D}$ of size $(B, C, H, W)$ and writes them to HBM. This step is unavoidable as the subsequent sampling requires random access to the spatial grid to perform adaptive sampling. To keep this I/O overhead small, we employ a lightweight state representation whose storage is on the same order as a standard feature map, i.e., ($O(BCL)$ rather than $O(BCNL)$), ensuring that the overhead remains negligible compared to the channel-expanded states of standard SSMs.

### B.0.2. MEMORY

The memory footprint of the Deformba block is optimized to avoid the quadratic memory bottlenecks associated with storing attention matrices. In our sample implementation, the states are realized in memory, incurring a $O(BCL)$ memory cost. The peak memory usage is dominated by the storage of the feature maps $\mathbf{T}$ and $\mathbf{V}$ of shape $(B, C, H, W)$. By avoiding the materialization of an $L \times L$ attention matrix, the IO memory requirement remains linear $O(L)$. The offset generation requires storing offset maps of size $(B, 2G, H, W)$, where the number of groups $G$ is smaller than the channel dimension $C$, ensuring the memory overhead for the adaptive sampling structure remains a small fraction of the total feature map size.

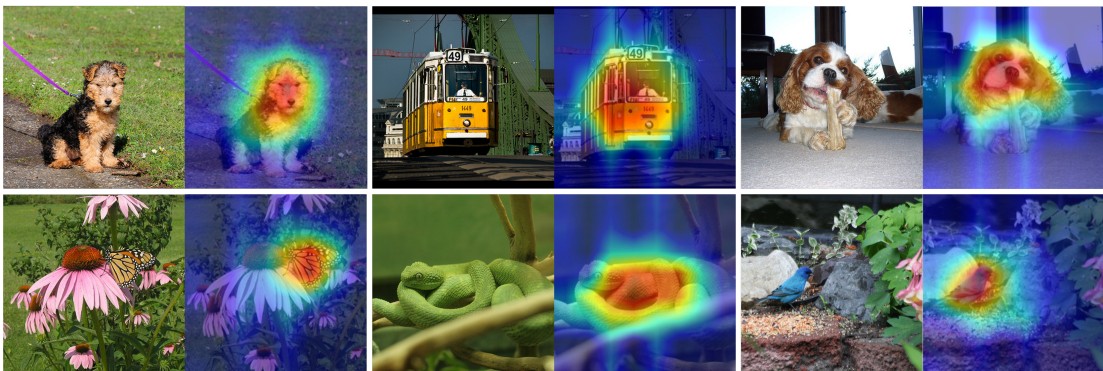

*Figure 6.* **Visualization of the last Deformba block in Stage 4 using Deformba-T.** For each example, we show the input image (left) and the corresponding activation/attribution map (right) from the final block of Stage 4. The model consistently highlights semantically discriminative regions while suppressing background, suggesting effective context aggregation by CASF.

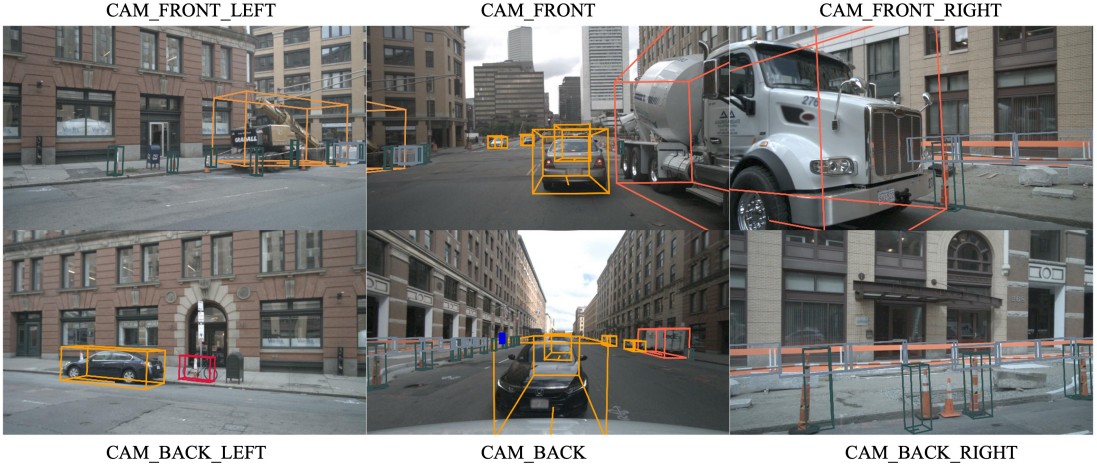

*Figure 7.* Visualization results of Deformba-Small for 3D bboxes predictions on nuScenes val set.

## B.1. Visualization and Algorithms

We provide grad cam figures for Deformba self attention in Figure 6 to show overall attention on input. For Deformba cross attention, we provide the 3D bboxes predictions in multi-camera images and the bird's-eye-view, as shown in Figure 7. The Pseudo code of our Deformba Self Attention are shown in Algorithm 1 and Cross Attention are shown in Algorithm 2.

## B.2. Additional Related Work

Beyond the vision SSM literature discussed in the main paper, recent works have explored efficient modeling (Wang et al., 2026; Jiang et al., 2026; Feng et al., 2026). Moreover, recent vision-language studies further investigate cross-modal representation fusion and multimodal reasoning (Liu et al., 2025b; Lin et al., 2026). These directions are complementary to our focus on state-space-based visual perception rather than vision-language reasoning.

## B.3. Limitations

Deformba should not be interpreted as only relying on CASF, as the architecture still benefits from additional locality priors including the convolutional components in each block. This suggests the gains come from the combination of compact state memory, adaptive readout, and local inductive bias.

---

**Algorithm 1** Deformba Block

---

**Require:** token sequence $\mathbf{T}_{l-1} : (\texttt{B}, \texttt{C}, \texttt{H}, \texttt{W})$
**Ensure:** token sequence $\mathbf{T}_l : (\texttt{B}, \texttt{C}, \texttt{H}, \texttt{W})$
1: $\mathbf{T}_{dw} : (\texttt{B}, \texttt{C}, \texttt{H}, \texttt{W}) \leftarrow \mathbf{DWConv2D}(\mathbf{T}_{l-1})$
2: $\mathbf{T}_{eca} : (\texttt{B}, \texttt{C}, \texttt{H}, \texttt{W}) \leftarrow \mathbf{T}_{dw} \odot \mathbf{Sigmoid}(\mathbf{Conv1d}(\mathbf{Mean}(\mathbf{T}_{dw})))$
3: $\tilde{\mathbf{V}}_{2D} : (\texttt{B}, \texttt{C}, \texttt{H}, \texttt{W}) \leftarrow \mathbf{Linear}^{in}(\mathbf{T}_{l-1})$
4: $\mathbf{V}_{2D} : (\texttt{B}, \texttt{C}, \texttt{H}, \texttt{W}) \leftarrow \mathbf{SiLU}(\mathbf{DWConv2D}(\tilde{\mathbf{V}}_{2D}) + \tilde{\mathbf{V}}_{2D})$
5: $\mathbf{V} : (\texttt{B}, \texttt{C}, \texttt{L}) \leftarrow \mathbf{Rearrange}(\mathbf{V}_{2D})$
6: $\mathbf{dt} : (\texttt{B}, \texttt{R}, \texttt{L}), \mathbf{K} : (\texttt{B}, \texttt{N}, \texttt{L}), \mathbf{Q} : (\texttt{B}, \texttt{N}, \texttt{L}) \leftarrow \mathbf{Split}(\mathbf{Linear}^V(\mathbf{V}))$
7: $\alpha : (\texttt{B}, \texttt{L}, \texttt{C}, \texttt{N}) \leftarrow \exp(\mathbf{Linear}^{dt}(\mathbf{dt}) \otimes \mathbf{A})$
8: $\mathbf{S}_0 : (\texttt{B}, \texttt{C}, \texttt{N}) \leftarrow \text{zeros}$
9: **for** $t$ in $1, ..., L$ **do**
10: $\quad \mathbf{S}_t : (\texttt{B}, \texttt{C}, \texttt{N}) = \alpha_t \mathbf{S}_{t-1} + \mathbf{V}_t \mathbf{K}_t^\top$
11: **end for**
12: $\mathbf{S}_{2D} : (\texttt{B}, \texttt{C}, \texttt{H}, \texttt{W}) \leftarrow \mathbf{Rearrange}([\mathbf{S}_1, ..., \mathbf{S}_L])$
13: $\mathbf{S}_Q : (\texttt{B}, \texttt{C}, \texttt{L}) \leftarrow \mathbf{CASF}(\mathbf{S}_{2D})$
14: $\mathbf{O} : (\texttt{B}, \texttt{C}, \texttt{L}) = \mathbf{S}_Q \odot \mathbf{Q} + \mathbf{V} \odot \mathbf{D}$
15: $\mathbf{T}_l : (\texttt{B}, \texttt{C}, \texttt{H}, \texttt{W}) \leftarrow \mathbf{Linear}^{out}(\mathbf{LayerNorm}(\mathbf{Rearrange}(\mathbf{O})))$
16: **Return** $\mathbf{T}_l$

---

**Algorithm 2** Deformba XA Block for BEV

---

**Require:** token sequence $\mathbf{T}_{l-1} : (\texttt{B}, \texttt{C}, \texttt{H}, \texttt{W})$, query sequence $\mathbf{M}_{l-1} : (\texttt{B}, \texttt{C}, \texttt{P})$
**Ensure:** query sequence $\mathbf{M}_l : (\texttt{B}, \texttt{C}, \texttt{P})$
1: $\tilde{\mathbf{V}}_{2D} : (\texttt{B}, \texttt{C}, \texttt{H}, \texttt{W}) \leftarrow \mathbf{Linear}^{in}(\mathbf{T}_{l-1})$
2: $\tilde{\mathbf{V}} : (\texttt{B}, \texttt{C}, \texttt{L}) \leftarrow \mathbf{Rearrange}(\tilde{\mathbf{V}}_{2D})$
3: $\mathbf{V} : (\texttt{B}, \texttt{C}, \texttt{H}, \texttt{W}) \leftarrow \mathbf{SiLU}(\mathbf{DWConv1D}(\tilde{\mathbf{V}}))$
4: $\mathbf{dt} : (\texttt{B}, \texttt{R}, \texttt{L}), \mathbf{K} : (\texttt{B}, \texttt{N}, \texttt{L}) \leftarrow \mathbf{Linear}^V(\mathbf{V})$
5: $\tilde{\mathbf{Q}} : (\texttt{B}, \texttt{N}, \texttt{P}), \mathbf{Z} : (\texttt{B}, \texttt{C}, \texttt{P}) \leftarrow \mathbf{Linear}^M(\mathbf{M}_{l-1})$
6: $\mathbf{Q} : (\texttt{B}, \texttt{C}, \texttt{P}) \leftarrow \mathbf{Linear}^Q(\mathbf{SiLU}(\tilde{\mathbf{Q}}))$
7: $\alpha : (\texttt{B}, \texttt{C}, \texttt{L}) \leftarrow \exp(\mathbf{Linear}^{dt}(\mathbf{dt}) \otimes \mathbf{A})$
8: $\mathbf{S}_0 : (\texttt{B}, \texttt{C}, \texttt{N}) \leftarrow \text{zeros}$
9: **for** $t$ in $1, ..., L$ **do**
10: $\quad \mathbf{S}_t : (\texttt{B}, \texttt{C}, \texttt{N}) = \alpha_t \mathbf{S}_{t-1} + \mathbf{V}_t \mathbf{K}_t^\top$
11: **end for**
12: $\mathbf{S} : (\texttt{B}, \texttt{C}, \texttt{L}) \leftarrow \mathbf{LayerNorm}(\mathbf{Rearrange}([\mathbf{S}_1, ..., \mathbf{S}_L]) + \mathbf{Rearrange}(\mathbf{V}))$
13: $\mathbf{offset} : (\texttt{B}, \texttt{P}, \texttt{E}, \texttt{F}, 2), \mathbf{weight} : (\texttt{B}, \texttt{P}, \texttt{F}) \leftarrow \mathbf{Linear}^{off}(\tilde{\mathbf{Q}})$
14: $\mathbf{S}_Q : (\texttt{B}, \texttt{C}, \texttt{P}) \leftarrow \mathbf{CASF}(\mathbf{S}, \mathbf{offset}, \mathbf{weight})$
15: $\mathbf{O} : (\texttt{B}, \texttt{C}, \texttt{P}) = \mathbf{S}_Q \odot \mathbf{Q} + \tilde{\mathbf{Q}} \odot \mathbf{D}$
16: $\mathbf{O}_{gated} : (\texttt{B}, \texttt{C}, \texttt{P}) \leftarrow \mathbf{LayerNorm}(\mathbf{O} \odot \mathbf{Z})$
17: $\mathbf{M}_l : (\texttt{B}, \texttt{C}, \texttt{P}) \leftarrow \mathbf{Linear}^{out}(\mathbf{O}_{gated})$
18: **Return** $\mathbf{M}_l$

---

