# OpenReview forum: "Deformba: Vision State Space Model with Adaptive State Fusion"
_ICML.cc/2026/Conference — ICML 2026 regular_

### Official Review · Reviewer_dSnq · 2026-03-07

**Soundness:** 3
**Presentation:** 3
**Significance:** 3
**Originality:** 2
**Overall Recommendation:** 3
**Confidence:** 2

**Summary:**

This paper identifies a limitation in Vision State Space Models where spatial interactions are constrained by a fixed scanning scheme. To address this issue, the authors reconstruct the state sequence produced by the scan into a 2D state map and propose a CASF mechanism that selectively aggregates spatially relevant states through dynamic offset-based sampling on this map. This approach enables context-adaptive spatial interaction while preserving the linear computational complexity of SSMs, and experiments demonstrate that the same structure can be applied to both self-interaction and cross-interaction.

**Compliance With Llm Reviewing Policy:**

Affirmed.

**Final Justification:**

Core concerns on spatial interpretability, offset learning validity, and lack of principled justification remain unresolved, so my evaluation remains unchanged.

**Key Questions For Authors:**

(1) Due to the issues mentioned in the Weaknesses, can the authors provide qualitative evidence showing that Deformba selects spatial information as intended? If so, can the authors theoretically explain whether this behavior arises from properties of SSMs (e.g., linearly separable features in $S_t$) or from other factors that may have been overlooked?

(2) Can the authors provide evidence that CASF enables semantic retrieval comparable to content-based cross-attention, rather than merely spatial sampling?

**Limitations:**

It would be interesting to analyze how the construction of $S_{2d}$ is affected as the complexity of the input image increases, and whether the influence of this complexity remains bounded.

**Strengths And Weaknesses:**

**Strengths**
- A clear and intuitively motivated problem formulation.
- Maintains linear complexity, thereby preserving the computational efficiency of SSMs.
- Extensive and empirical evaluations across diverse tasks, including image classification, object detection, semantic segmentation, and BEV perception.
- A conceptual design that interprets the SSM state as a memory and separates sequence updating (write) from information retrieval (read).
- Broad applicability, supporting both self-attention and cross-attention style interactions.

**Weaknesses**
- The use of offsets follows existing designs and does not introduce mechanisms specifically tailored to SSMs, which limits the originality of the approach.
- Eq.(1) follows a prefix accumulation structure, each position in the state map is not guaranteed to represent information from the corresponding spatial location. Therefore, it is unclear whether CASF sampling truly performs spatially localized evidence retrieval.
- CASF claims to learn spatial relevance through offset prediction, the model relies solely on the downstream loss without auxiliary signals such as contrastive learning, correspondence supervision, or masked modeling. As a result, it remains unclear whether the learned offsets truly correspond to meaningful spatial sampling.
- The author claims that CASF can perform query-to-context retrieval similar to cross-attention. However, the mechanism relies on spatial offset-based sampling rather than content-based similarity, leaving it unclear to what extent it can truly substitute the functionality of cross-attention.

Writing
- A typo appears in line 255 ("and and")
- In Eq. 2, the variables $O, Q$ are not defined at the first place.
- In Table 4, for the Deformba-S setting with Frames = 3, is the bold value for NDS a typo, or does it indicate the best score within the frame settings?

---

> ### Author Rebuttal · Authors · 2026-03-31
>
> Dear Reviewer,
>
> Thank you for your insightful review and feedback. We are delighted that you found our novel conceptual design. Below, we address your concerns and comments in detail:
>
> **W1: The use of offsets follows existing designs and does not introduce mechanisms specifically tailored to SSMs, which limits the originality of the approach.**
>
> Sorry for the confusion! We respectfully clarify that our contribution is not the offset operator in isolation, but the SSM-specific write-read decoupled formulation that makes offset based retrieval meaningful in the first place. Prior vision SSMs typically entangle memory writing and retrieval within fixed scans, whereas CASF treats the scan as a memory writing stage and introduces adaptive retrieval over the resulting state memory, rather than to raw feature aggregation. This write-read decoupling is what enables the same mechanism to support both self-mixing and query-to-context fusion.
>
> This design is tailored to a specific limitation of vision SSMs: the write process is causal and scan-dependent, while visual interaction is often non-causal and, in cross-query settings, query-dependent. CASF therefore uses offsets as a mechanism for adaptive memory access over the written state map, instead of treating them as a generic deformable feature operator. We will revise the manuscript to make this SSM-specific motivation and distinction clearer.
>
> **W2: Prefix accumulation may weaken location semantics.**
>
> We agree that, due to prefix accumulation in Eq. (1), the location in ​$S_{2D}$ should not be interpreted as a purely local feature corresponding only to the same spatial position. A more precise interpretation is that ​$S_{2D}$ is a spatially indexed contextual memory, where each slot remains anchored to a spatial index after reshaping, but already contains accumulated context from the recurrent write process. Therefore, CASF should be understood as spatially indexed memory retrieval, rather than strictly local evidence retrieval. We will revise the manuscript to clarify this distinction.
>
> Accompany with questions in **Q1: qualitative evidence** and **Q2: semantic retrieval evidence**, we provide qualitative evidence for this behavior by visualizing the learned sampling locations. In our visualization (find in the link below), the offsets do not collapse to zero or a fixed pattern; instead, they spread to different semantic places around the reference point in an input-dependent manner. This shows our CASF learns non-trivial adaptive retrieval over the written state map.
>
> For another concern in **Q1: theoretically explain**, theoretically, we do not attribute this behavior to a strong assumption such as mentioned $S_{t}$ is linearly separable. We think it arises from three factors: (i) the spatial anchoring of the written state map after reshaping, (ii) the broader contextual conditioning available in $S_{2D}$ for offset prediction, and (iii) the differentiable sampler through downstream task gradients directly optimize which memory slots are most useful to read. So that the learned offsets are best interpreted as task-driven memory access over SSM state memory.
>
>
> **W3: No auxiliary supervision for offsets.**
>
> Thank you for your question. Our current method learns offsets from the downstream task loss. We provide grad cam figures in figure 5 to show overall attention on input, but it does not explain what the spatial sampling is doing to get to that figure. To better support the effectiveness of our method, we also provide offset visualizations and visualization of sampling distribution in the anonymous link below.
>
> **W4: Similar to cross-attention may be too strong**, and another concern in **Q2: If CASF enables semantic retrieval comparable to content-based cross-attention, rather than merely spatial sampling**.
>
> We agree that CASF is not a strict replacement for standard content-based cross-attention. A more accurate description is that CASF performs query-specific retrieval over the written state memory. The retrieval locations are not fixed, but are predicted from the current token, so the mechanism goes beyond predefined spatial sampling. We will revise the manuscript to avoid overstating equivalence to dot-product content attention.
>
> **Writing issue:**
>
> 1. A typo appears in line 255 ("and and").
>
> 2. In Eq. 2, the variables $O, Q$ are not defined at the first place.
>
> $Q \in R^{B,L,N}$ is projection parameter for input sequences to decode the state. $O\in R^{B,C,N}$ is the output.
>
> 1,2: Thank you for your careful reading of our paper. We have fixed the writing issues you highlighted in the revised paper.
>
> 3. In Table 4, for the Deformba-S setting with Frames = 3, is the bold value for NDS a typo, or does it indicate the best score within the frame settings?
>
> Sorry for confusion. Yes, it indicate the best score within the frame settings.
>
> [anonymous link](https://anonymous.4open.science/r/Deformba-Vision-State-Space-Model-with-Adaptive-State-Fusion-BAB4)

---

> > ### Author Rebuttal · Reviewer_dSnq · 2026-04-01
> >
> > Thank you for the detailed and thoughtful responses. While the authors provide helpful clarifications by reframing their contribution as a write–read decoupled design and appropriately toning down the claim regarding cross-attention equivalence, the core concerns remain insufficiently addressed. In particular, although the authors acknowledge that the state map should be interpreted as a contextual memory rather than a strictly local feature, it is still unclear whether such a representation supports spatially meaningful retrieval, given the prefix accumulation structure in Eq. (1). Furthermore, the offset prediction mechanism continues to rely solely on downstream supervision, with limited evidence that it learns genuinely meaningful sampling beyond heuristic behavior. Finally, while the claim is softened, the gap between spatial sampling and content-based retrieval is not theoretically or empirically resolved. As a result, the rebuttal improves clarity but does not provide sufficient justification to alter the original assessment.

---

> > > ### Author Response · Authors · 2026-04-06
> > >
> > > Thank you for the follow-up. Below are our detailed responses to your question.
> > >
> > > >**It is still unclear whether such a representation supports spatially meaningful retrieval, given the prefix accumulation structure in Eq. (1)**
> > >
> > > We would like to clarify that, in addition to the attribution (grad_cam) visualization in the paper (Fig. 5), we have also provided supplementary visualizations in the anonymized repository, which we attached the link before,
> > > ([https://anonymous.4open.science/r/Deformba-Vision-State-Space-Model-with-Adaptive-State-Fusion-BAB4/README.md](https://anonymous.4open.science/r/Deformba-Vision-State-Space-Model-with-Adaptive-State-Fusion-BAB4/README.md)) including:
> > >
> > > (1) an ERF comparison showing that predicting offsets from the written state map yields a broader effective receptive field than predicting offsets directly from raw features.
> > >
> > > (2) offset visualization and sampling distribution visualization. These provide evidence of semantically grounded sampling, since the sampled locations tend to fall on the same target object and nearby semantically related regions which shows the learned sampling is non-trivial, input-conditioned and spatially aligned with semantically relevant object regions. These visualizations also provide evidence that CASF learns non-degenerate adaptive sampling.
> > >
> > > The figures and statistical distribution should supports our method can truly perform the spatially meaningful retrieval in the state map.
> > >
> > > For you another concern about prefix accumulation structure in SSM, the state can be thought of as accumulation of the KV outer product (in the language of dot product attention alternatively Bx in the language of mamba). In a unidirectional scan that information is accumulated causally across a flattened image. The operation we do on state space is to build the connectivity between current feature/query and its related neighborhoods at state level. Specifically, by introducing sampling points which can 1. retrieve information from future states and 2. do so by moving sampling points across a 2D feature map we reintroduce the geometric biases which best suit vision tasks and break the causal constraint. In another view, this allows us to better approximate full, non-causal, scaled dot product attention by leveraging this geometric bias to select future states which are most likely to have a high dot product similarity with the query.
> > >
> > > Moreover, Deformba is not relying on CASF alone; the architecture still benefits from additional locality priors such as the convolutional components in the block, which suggests that its gains come from the combination of compact state memory, adaptive readout, and local inductive bias.
> > >
> > > >**Furthermore, the offset prediction mechanism continues to rely solely on downstream supervision, with limited evidence that it learns genuinely meaningful sampling beyond heuristic behavior.**
> > >
> > > We respectfully note that learning the offsets solely from downstream supervision is a standard and reasonable design choice for differentiable spatial sampling modules. This is not a limitation of in itself. In such modules, the key question is not whether auxiliary supervision is present, but whether the learned behavior is non-degenerate and task-relevant. In our case, the learned offsets are optimized end-to-end through differentiable sampling, and our supplementary visualizations (the attached link in our first rebuttal section and above) show that they do not collapse to zero offsets or a fixed predefined pattern, but vary with the input/query and often align with semantically relevant object regions.
> > >
> > > >**The claim for our paper**
> > >
> > > The claim of our paper is: our CASF decouples SSM's write and read, so the same mechanism supports both **self-attention-style mixing** and **cross-attention-style query-to-source retrieval** with minimal changes to the write pass. We believe this write-once, adaptive-read-many paradigm is the key conceptual novelty of our work.
> > >
> > > For you concern about cross-attention-style query-to-source retrieval, our query-to-source interaction mechanism is based on predicted spatial offsets over written state memory but not explicit query-key similarity. And we clarify in paper that the cross attention specifically indicates query-to-source interaction. To avoid misunderstanding, in the revision we will avoid wording that suggests equivalence to standard cross-attention, and instead describe CASF as an efficient query-to-source interaction mechanism, which is practically useful for both self-mixing and query-to-context fusion within the SSM framework.
> > >
> > > Thank you again for your valuable feedback. We would be very grateful if you could take our additional evidence into account when making your final assessment.

---

### Official Review · Reviewer_3KcA · 2026-03-08

**Soundness:** 3
**Presentation:** 3
**Significance:** 3
**Originality:** 4
**Overall Recommendation:** 5
**Confidence:** 4

**Summary:**

This paper introduces Deformba, a novel Vision State Space Model (SSM) designed to overcome the limitations of fixed scanning paths and the lack of native cross-attention mechanisms in existing vision SSMs. The core contribution is the Context-Adaptive State Fusion (CASF) mechanism, which decouples the SSM's "write" and "read" operations. By using a single unidirectional scan to build a shared 2D state memory and employing learnable offsets to dynamically sample spatially relevant features, Deformba effectively models complex 2D spatial contexts without relying on predefined geometric orders. Furthermore, this decoupled design naturally supports query-to-source interactions (analogous to cross-attention) while maintaining the linear complexity of SSMs. Extensive experiments demonstrate its strong and versatile performance across standard 2D tasks (classification, detection, segmentation) and 3D multi-view BEV perception.

**Compliance With Llm Reviewing Policy:**

Affirmed.

**Final Justification:**

I appreciate the authors’ efforts during the rebuttal period. The responses satisfactorily address all of my concerns.

**Key Questions For Authors:**

see weakness

**Limitations:**

yes

**Strengths And Weaknesses:**

Strengths*

- the paper introduces a highly creative decoupling of the State Space Model's (SSM) "write" and "read" operations. By abandoning rigid, predefined scanning paths and instead using a single scan to build a 2D state memory, the model dynamically samples features using context-adaptive offsets (CASF). This successfully frees vision SSMs from restrictive 1D causal assumptions.

- The work addresses a critical bottleneck in the vision SSM literature: the lack of a native mechanism for query-to-source interaction. By enabling cross-attention-style fusion within an SSM framework, Deformba unlocks new directions for SSM applications in complex multi-stream perception tasks, such as multi-view 3D BEV detection.

- The submission is exceptionally well-structured and clearly written. It effectively positions itself within the current literature by clearly identifying the two main flaws of existing vision SSMs (fixed geometric scanning and inability to handle multi-stream queries). The transition from identifying these problems to introducing the CASF solution is logical and easy to follow.


Major Weaknesses*

- The paper repeatedly emphasizes its $\mathcal{O}(L)$ linear complexity and high architectural efficiency, but only reports theoretical parameters and FLOPs in all major experimental tables (Tables 1-5). For such fundamental architectural improvements—especially those introducing memory-bound operations like bilinear interpolation—it is crucial to provide actual inference latency or throughput (FPS/IPS) comparisons on physical GPUs. The absence of these hardware-level metrics severely undermines the claims of "high efficiency."

- The core mechanism is termed "Context-Adaptive State Fusion (CASF)" and claims to decouple the read/write operations of SSMs. However, as detailed in Appendix B, the model explicitly materializes a 2D feature map $S_{2D}$ of shape $(B, C, H, W)$ into High Bandwidth Memory (HBM). This is essentially the output feature map after the SSM scan. Performing bilinear interpolation with learned offsets on this output map is mathematically nearly identical to Deformable Convolution (DCN) or Deformable Attention. Over-packaging this established mechanism as a revolutionary operation on the "internal hidden state" of an SSM is academically misleading.

- Since the core of CASF relies on learning offsets for spatial feature sampling, it is imperative to compare Deformba against advanced baseline models that utilize similar deformation mechanisms (particularly the DCNv3-based InternImage family) in object detection (Table 2) and semantic segmentation (Table 3). Comparing solely against Swin, ConvNeXt, and early VMamba variants is insufficient to demonstrate Deformba's absolute superiority in the "deformable/adaptive sampling" track.

- The paper specifically introduces the Efficient Channel Attention (ECA) module into the offset prediction network to suppress inherent channel redundancy in Mamba architectures. However, the ablation study in Table 6 shows that CASF w/o ECA achieves 81.3% Top-1 accuracy, while CASF w/ ECA yields 81.4%. On a dataset of ImageNet's scale, a mere 0.1% improvement is typically considered statistical noise. The authors fail to quantitatively justify the absolute necessity of introducing this attention module.

- A core argument of the paper is abandoning complex multi-directional scanning in favor of a basic single-directional scan. The ablation in Table 7 shows that single and bidirectional scans yield identical mAP scores (0.393). While the empirical data supports this simplification, the authors provide absolutely no theoretical explanation—from the perspectives of receptive fields, state compression, or spatial information flow—as to why a single causal SSM scan followed by deformable sampling is perfectly adequate for strictly non-causal 2D image data.

---

> ### Author Rebuttal · Authors · 2026-03-31
>
> Dear Reviewer,
>
> Thank you for your valuable feedback, and the time taken to provide constructive feedback to strengthen our work further! We hope our following responses can address your concerns.
>
> **W1:  Actual inference latency or throughput (FPS/IPS).**
>
> Thank you for your comment, we agree that only reports parameters and FLOPs is insufficient. We add the trade-off between ImageNet-1K top-1 accuracy and inference throughput. All the models are trained under the DeiT training hyperparameters. Actual hardware used for inference throughput is an NVIDIA RTX 6000 Ada GPU with a batch size 128.
>
> | Model |Inference Throughput (image/sec) |Top-1 acc. (%) |
> |:--|:--:|:--:|
> |DefMamba-S|581.0|83.5|
> |DefMamba-B|331.7|84.2|
> |TransNeXt-T|425.4|84.0|
> |TransNeXt-S|220.6|84.7|
> |TransNeXt-B|152.4|84.8|
> |ConvNeXt-T|1158.8|82.5|
> |ConvNeXt-S|677.7|83.1|
> |ConvNeXt-B|468.1|83.8|
> |**Deformba-T**|993.4|83.8 |
> |**Deformba-S**| 531.3   |  84.9 |
> |**Deformba-B**|  398.8  |   85.4 |
>
> It can be seen that under the same inference throughput or accuracy, the accuracy or inference throughput of our proposed Deformba outperforms the SSMs(DefMamba), ViTs(TransNeXt) and CNNs(ConvNeXt) type's model.  We also provide the figure in link below.
>
> **W2: "The core mechanism is termed "Context-Adaptive State Fusion (CASF)" and claims to decouple the read/write operations of SSMs..."**
>
> We agree that the current wording can be misread as over-packaging, and we will revise it. Our claim is not that bilinear interpolation itself is new. The key distinction is that CASF samples from a written recurrent state memory, after decoupling write and read, rather than directly from raw current features. This difference is what enables the same operator to serve both self-mixing and external query-conditioned retrieval. We also note that Appendix B of our manuscript already explicitly acknowledges that $S_{2D}$ is materialized in HBM because random spatial access is required; so we are not claiming a hidden-state implementation trick that avoids materialization. And we also provide the visualization of effective receptive field (ERF) for the offset prediction module to show our method can obtain larger ERF than deformable attention.
>
> **W3: Comparison to InternImage / DCNv3 family.**
>
> This is a good suggestion. InternImage explicitly uses DCNv3 as its core operator for adaptive spatial aggregation, so it is indeed a relevant baseline family for the deformable/adaptive sampling axis.
>
> | Model | 	resolution |  Params(M) |  FLOPs(G) 	|  Top-1 acc. (%) |
> |:--|:--:|:--:|:--:|:--:|
> |InternImage-T| 224x224 | 30  |  5  |   83.5  |
> |**Deformba-T**| 224x224  |  25  | 4.8  |   83.8 |
> |InternImage-S|  224x224 |  50  |  8 | 84.2   |
> |**Deformba-S**| 224x224 | 45  |   10.3 |   84.9 |
> |InternImage-B| 224x224 |  97 |   16 |  84.9  |
> |**Deformba-B**|  224x224 | 85   |  16.3  |   85.4 |
>
> Our Deformba consistently outperform InternImage. We will add this comparison in the revised paper.
>
> **W4: ECA only gives +0.1.**
>
> Thank you for your question. We agree that the gain from ECA is modest. We have this module because considering the previous method’s findings[1, 2] that mamba has redundancy in the channel dimension,  ECA is used for channel recalibration. Our intention was not to present ECA as a major contributor, but as a lightweight channel calibration module with negligible k learnable parameters (where k is the 1D kernel size). Therefore, its computational overhead is negligible and ECA should be viewed as an optional refinement, while CASF remains the main source of improvement.
>
> **W5: Why is a single scan enough?**
>
> Thank you for your question. We agree that the current draft lacks a more explicit theoretical explanation. In Deformba, the scan is used to construct a state memory $S_{2D}$, while the final interaction topology is established by CASF through adaptive reading. Formally, the write recurrence $S_{t} = \alpha S_{t-1} + v_{t}k_{t}^{T}$, defines a fixed directed write graph, whereas CASF introduces learned read edges via equation 5 and 6 in paper. Hence, the effective interaction graph is $G_{eff} = G_{write} \cup G_{read}$, rather than being determined by scan order alone.
>
> This also explains why single scan can be adequate in practice. The final receptive field of token i becomes a union of multiple written state receptive fields selected by adaptive reading. From the perspective of state compression, this avoids forcing the write process alone to preserve all 2D dependencies, which would otherwise require a larger state. In this sense, the role of the single scan is efficiency-preserving state construction, while CASF provides the missing non-causal spatial information flow.
>
> [anonymous link](https://anonymous.4open.science/r/Deformba-Vision-State-Space-Model-with-Adaptive-State-Fusion-BAB4)
>
> [1]Groupmamba: Parameter-efficient and accurate group visual state space model.
>
> [2]Multi-scale vmamba: Hierarchy in hierarchy visual state space model.

---

> > ### Author Rebuttal · Reviewer_3KcA · 2026-04-01
> >
> > I appreciate the authors’ efforts during the rebuttal period. The responses satisfactorily address all of my concerns.

---

> > > ### Author Response · Authors · 2026-04-06
> > >
> > > We sincerely thank this reviewer for the positive feedback and for accepting our work!

---

### Official Review · Reviewer_jyQX · 2026-03-13

**Soundness:** 3
**Presentation:** 2
**Significance:** 3
**Originality:** 2
**Overall Recommendation:** 3
**Confidence:** 3

**Summary:**

This paper proposes Deformba, a vision State Space Model (SSM) designed to overcome the limitations of fixed scanning patterns and the lack of query-based fusion mechanisms in existing vision SSMs. The core contribution is Context-Adaptive State Fusion (CASF), which decouples SSM write and read operations: a single scan populates a 2D state memory, while an offset prediction network enables adaptive sampling of spatial evidence. This unified design supports both self-attention-style feature mixing and cross-attention-style query-to-source interaction, achieving competitive results across 2D classification, detection, segmentation, and 3D BEV perception tasks.

**Compliance With Llm Reviewing Policy:**

Affirmed.

**Final Justification:**

While I recognize the empirical performance advantages demonstrated by Deformba and do not oppose its acceptance based on its practical value, there are two significant logical and structural flaws in the manuscript and rebuttal that must be clearly addressed:

1. The theoretical justification for the "single scan" remains untenable:
Throughout the two rounds of rebuttal, the authors have argued that the single sweep scan serves merely as a "write" pass (analogous to KV accumulation in Transformers) , suggesting that the subsequent adaptive "read" pass can overcome directional bias. However, this analogy is fundamentally flawed. Unlike the lossless attention matrices in Transformers, the intrinsic mechanism of State Space Models (SSMs) relies on the lossy compression of hidden states. During the 1D serialization of 2D images, features that are spatially adjacent but sequentially distant inevitably suffer from severe attenuation , imprinting an irreversible directional bias onto the base state map ($S\_{2D}$) . No matter how flexible the CASF (Context-Adaptive State Fusion) module’s reading mechanism is, it cannot recover spatial structural information that was already degraded or lost during the writing phase.

2. Reliance on external links and missing methodological details:
While I understand the challenges posed by the 5000-character rebuttal limit, providing critical clarification via an external anonymous link deviates from standard policies. Beyond this procedural note, this situation highlights another issue: these core methodological details, which are necessary for fully understanding the model architecture, were missing from the original manuscript.

If this paper is ultimately accepted, it is imperative that the authors explicitly acknowledge the theoretical limitations of single-direction scanning in the Camera-ready version and incorporate all missing core mathematical derivations into the main text.

**Key Questions For Authors:**

1. The authors explicitly state in the introduction that existing vision SSMs rely on fixed scanning paths to flatten 2D features into 1D sequences, which imposes predefined geometric structures and destroys spatial continuity. However, in the proposed Context-Adaptive State Fusion (CASF), the authors still employ a "single sweeping scan" to generate the 2D state map $S_{2D}$. This presents a fundamental logical contradiction: the foundational state map generated by a single scan inherently suffers from strong directional bias. Even with subsequent adaptive sampling, the base features are already compromised by this bias. The authors fail to demonstrate or justify that a single scan is sufficient to produce a high-quality, unbiased $S_{2D}$.

2. The proposed feature offset prediction network utilizes Efficient Channel Attention (ECA) and attempts to inject global context via Global Average Pooling. The Context-Adaptive State Fusion (CASF) mechanism heavily relies on precise global spatial awareness. Simply using average pooling to compress a 2D feature map that possesses high spatial diversity is arguably too coarse of a strategy. It is highly questionable whether this approach can extract sufficiently fine-grained spatial reasoning cues to accurately guide the generation of dense offset maps.

3. Equation (3) indicates the generation of $G$ groups of 2D offsets , yet neither the main text nor the experimental settings clearly specify the exact value of $G$ or its configuration rationale. Furthermore, computing the augmented state utilizes the sampling point coordinates $\mathcal{P}=\mathcal{E}+\Delta p$ . However, the specific source, generation method, and initialization strategy for the reference point coordinates $\mathcal{E}$ are completely missing from the manuscript.

4. The paper introduces learnable weights $w_g$ and uses a summation operation to generate the final query state $S_Q$ . However, the previously described $f_{off}(\cdot)$ network only outputs the spatial offsets $\Delta p$ , leaving it entirely unexplained how the weights $w_g$ are predicted or calculated. Additionally, the authors need to justify the use of a simple summation aggregation. Compared to standard fractional weighted aggregation (as seen in Deformable Attention), simple summation carries a significant risk of diluting critical features.

5. The description of the Cross-Attention adaptation in Section 4.3 is overly brief and lacks the necessary mathematical formulation . This is particularly problematic for core tasks like multi-view 3D BEV detection. In such contexts, a critical step is how external queries utilize camera intrinsic and extrinsic matrices to project onto 2D feature maps to obtain initial reference points. This vital 3D-to-2D mapping process—essential for 3D perception—is entirely omitted.

6. The representation of multi-scale and multi-view features in Figure 3 is ambiguous . The figure illustrates features arranged as 1D sequences, but the text fails to explain the specific serialization or sorting rules applied to multi-view and multi-scale features when performing Context-Adaptive State Fusion. Moreover, it remains unclear how the model distinguishes and maintains spatial correspondences across different scales after such serialization.

7. The paper claims to achieve SOTA performance across multiple tasks, including classification, detection, segmentation, and BEV perception . However, no open-source code link (or anonymous repository) is provided. Given that the performance of SSMs in visual tasks heavily depends on low-level CUDA operator optimizations and memory management details (as acknowledged in Section 4.4 ), the absence of code makes it impossible for reviewers to verify the authenticity of the experimental results and the actual deployment efficiency of the proposed method.

8. The ablation studies presented in the experimental section are overly weak. Both the main text and the appendix (Tables 6 to 11) only ablate conventional hyperparameters (e.g., FFN channels, expansion factors, convolution types) . There is a lack of targeted and convincing ablation evidence for the core parameters and designs of this architecture.

**Limitations:**

yes.

**Strengths And Weaknesses:**

Strengths:

The model's primary strength lies in its ability to extend the SSM paradigm to non-causal spatial interaction and multi-modal fusion while maintaining linear computational complexity. By decoupling state materialization from sampling, Deformba provides a viable SSM-based alternative to cross-attention for complex perception tasks such as BEV representation learning. Furthermore, the architecture demonstrates strong empirical performance and scalability across multiple benchmarks, outperforming several state-of-the-art CNN, Transformer, and Mamba-based backbones with comparable parameter counts and FLOPs.

---

Weaknesses:

While the proposed architecture shows promise, there are concerns regarding the logical consistency of its scanning mechanism and the technical clarity of its mathematical formulations. Detailed critiques and specific technical ambiguities are further elaborated upon in the "Key Questions For Authors" section.

---

> ### Author Rebuttal · Authors · 2026-03-31
>
> Dear Reviewer,
>
> We sincerely thank you for your constructive comments! Please find our response to your concerns below:
>
> **Q1: Single scan vs. criticism of fixed scan paths.**
>
> Thank you for the important comment. We agree that a single sweep scan is not unbiased in a strict sense, and our intention is not to claim otherwise. Our point is that the limitation of prior vision SSMs is not only using a fixed scan, but letting the scan path itself determine the final interaction topology. In our Deformba, the single scan is used only for a compact write pass that constructs a shared state memory $S_{2D}$, which flatten 2D images to 1D sequences so that they can be processed by SSM. The final spatial interaction is then established by CASF through adaptive reading from the written state map.
>
> Therefore CASF reduces the dependence of spatial modeling on scan direction by augmenting the fixed write graph with learned read connections. This is consistent with our original ablation in Table 8, where single and bidirectional traversals achieve the same mAP (0.393). To further verify this point, we additionally evaluate different scan patterns on ImageNet-1K using Deformba-T (following the scan settings of VMamba):
>
> Table1
>
> | Method | 	  Top-1 acc. |
> |:--|:--:|
> |Unidi-Scan|  81.33|
> |Bidi-Scan|  81.24 |
> |Cascade-Scan Row and Col|   81.28 |
> |Cross Scan|   81.38|
>
> The results are highly similar across different scan patterns. This suggests that, once CASF is introduced, the model becomes insensitive to specific scanning strategy, and the spatial interaction is governed mainly by adaptive reading rather than by the predefined scan order.
>
> **Q2: Too coarse for offset prediction.**
>
> Thank you for your question. Considering the previous method’s findings[1, 2] that mamba has redundancy in the channel dimension,  ECA is used for channel recalibration, not the sole source of spatial reasoning. The dense offsets are predicted from the full spatial state map $S_{2D}$; The global average pooling is used to provide low-cost global channel calibration with negligible k learnable parameters (where k is the 1D kernel size). Importantly, CASF without ECA already improves performance over the baseline (Table 6), while ECA adds only a small extra gain.
>
> **Q3: Definitions of G and E.**
>
> We apologize for the definitions of $G$ and $\mathcal{E}$. $G$ denotes the number of sampling groups and is defined as $\frac{d_{model}}{d_{head}}$, which is equivalently, heads. Therefore, $G$ is not an independently tuned hyperparameter, but is induced by the channel partition used in the module. In our current set, $G = 8$. The source of the reference point $\mathcal{E}$ is centered on the image feature query in self attention case. For cross attention in BEV perception case, the reference point $\mathcal{E}$ follows BEVFormer style initialization.
>
> **Q4: description of $w_g$.**
>
> We agree that the current description in line 226-227 that "...update the current state along with learned weights $w$" is not clear enough. In our implementation, the aggregation weights $w_g$ are also dynamically predicted from the input features. Concretely, the offset prediction branch outputs both $\Delta p$ and the corresponding $w_g$ for adaptive fusion. This design allows the model to adaptively emphasize informative sampled locations and suppress less useful ones, thereby mitigating the feature dilution issue raised by the reviewer. We will revise it in paper.
>
>
> **Q5 and Q6: BEV Cross Attention adaptation.**
>
> We sincerely thank you for the recognition. Due to the word limitation, the details for BEV Cross Attention adaptation which includes cross-attention and BEV formulation and multi-scale serialization is provided in the link below.
>
> **Q7: No code release.**
>
> Thank you for your interest. Due to policy limitation, full source code will be available on publication. But we provide pseudocode in link.
>
> **Q8: Targeted ablations are weak.**
>
> Thanks for the suggestion. To fully isolate the isolate which part of CASF is responsible for the gain, we add ablations:
>
> 1. replace deformable sampling with predefined non-adaptive alternatives and MLP. These indicate the improvement comes specifically from adaptive deformable state reading.
>
> | Method | Im. size | Param. (M) | FLOPs (G)	 | Top-1 acc. |
> |:--|:--|:--:|:--:|:--:|
> |zero-offset| 224x224 | 25.0 | 4.7  | 80.8|
> |fixed offset|  224x224| 25.0 | 4.8  | 80.9 |
> |MLP|  224x224| 30.8 | 5.6  |81.0 |
> |dynamic offset|  224x224| 25.5 | 4.8 | 81.3|
>
> 2. different scan patterns, the result shows on Table 1 above.
>
> 3. analysis of sampling behavior by Effective Receptive Field (ERF).
>
> 4. trade-off of accuracy and inference throughput.
>
> 3,4 results are in the link.
>
> [anonymous link](https://anonymous.4open.science/r/Deformba-Vision-State-Space-Model-with-Adaptive-State-Fusion-BAB4)
>
> [1]Groupmamba: Parameter-efficient and accurate group visual state space model.
>
> [2]Multi-scale vmamba: Hierarchy in hierarchy visual state space model.

---

> > ### Author Rebuttal · Reviewer_jyQX · 2026-04-01
> >
> > Thank you for your detailed and comprehensive rebuttal. Please find my further feedback regarding your responses below:
> >
> > 1. I appreciate the authors' explanation that the CASF module is designed to reduce directional bias. While I fully acknowledge the novelty of CASF in establishing dynamic read connections, my primary concern lies in the construction of the initial input. As the authors profoundly pointed out at the end of the second paragraph in the Introduction, any fixed scanning path inevitably perturbs the native 2D neighborhood structure, breaks spatial continuity, and introduces directional biases. Since the authors themselves explicitly acknowledge that the serialization process fundamentally disrupts the original 2D spatial structure, how can the subsequent CASF module—no matter how adaptive it is—logically recover and derive a perfectly accurate spatial representation from a foundational feature map ($S_{2D}$) that has already been irreversibly "perturbed"? Although you provided empirical results (e.g., Table 1 in the rebuttal) demonstrating that the final performance is relatively insensitive to the scanning pattern, this does not address my concern at a theoretical and logical level: how can an underlying state map with inherent geometric biases theoretically justify the soundness of the entire spatial modeling architecture?
> >
> > 2. The explanations provided in the rebuttal regarding the number of sampling groups $G$, the initialization strategy for the reference point $\mathcal{E}$, and the dynamic prediction mechanism for the aggregation weights $w_g$ are clear and reasonable. Please ensure that these crucial mathematical definitions, parameter derivations, and implementation details are thoroughly and accurately incorporated into the main text of the final version to guarantee the paper's rigor and complete reproducibility.
> >
> > 3. **On the Compliance of the External Anonymous Link:** I noticed that the detailed answers regarding the Cross-Attention adaptation (Q5, Q6) and parts of the ablation studies (Q8) were provided via an external anonymous link. According to the standard ICML rebuttal policies, external links are strictly restricted to hosting additional figures and tables to maintain fairness regarding word limits. While I am willing to be lenient and treat a reasonable amount of text surrounding the images as extended captions, I regret to note that the link contains substantial content where large blocks of PDF text have been directly screenshot and uploaded as "images." This practice essentially bypasses the length constraints and violates the fundamental fairness principles. Therefore, out of respect for and maintenance of these rules, I must regrettably treat the specific questions addressed via these non-compliant materials (primarily Q5 and Q6) as unanswered.

---

> > > ### Author Response · Authors · 2026-04-05
> > >
> > > Thank you for your response and acknowledge the novelty of our paper.
> > >
> > > **Q1: Theoretical justification**
> > >
> > > The role of single scan in Deformba is only to perform an efficient SSM write and produce a shared context-aware state memory; the actual interaction topology is no longer tied to that scan path, but is learned during the subsequent read stage through spatially adaptive sampling.
> > >
> > > Moreover, Deformba is not relying on CASF alone; the architecture also benefits from additional locality priors such as the convolutional FFN and convolutional positional encoding, which are standard modules in popular vision models.
> > >
> > > We also provide theoretical justification through an interpretation from the perspective of transformers: The state can be thought of as accumulation of the KV outer product (in the language of dot product attention alternatively Bx in the language of mamba). In a unidirectional scan that information is accumulated causally across a flattened image. By introducing sampling points which can 1. retrieve information from future states and 2. do so by moving sampling points across a 2D feature map we reintroduce the geometric biases which best suit vision tasks and break the causal constraint. In another view, this allows us to better approximate full, non-causal, scaled dot product attention by leveraging this geometric bias to select future states which are most likely to have a high dot product similarity with the query.
> > >
> > > **Response 2**
> > >
> > > We will include all mathematical definitions, parameter derivations, and implementation details in the final paper.
> > >
> > > **Q3: BEV Cross Attention adaptation**
> > >
> > > Thank you for pointing out the anonymous link. Because the rebuttal was limited to 5000 characters, we could not address Q5 and Q6 there; we provide the response below.
> > >
> > > Given a set of input image feature maps $F$ and a set of queries $B$, the goal is to learn a function $f$ which selects information from $F$ to update $B$. This consists of three steps: 1) construction of hidden states of $F$, 2) construction of hidden states of $B$, and 3) decoding of hidden states of $B$.
> > >
> > > Following [1], [2], [3], and [4] the encoding function $f_{enc}$ includes ResNet backbone, FPN, BEV encoder, and temporal fusion modules. First, multi-scale image features $F_0, ..., F_i$ are extracted from the backbone. Second, the FPN reduces them to a common channel dimension $D$ [5]. Then, we model feature interactions with our SSM-based pipeline. Lastly, the decoding function $f_{dec}$ operates on this representation to make predictions. We replace the cross attention module in encoder, we next discuss the details.
> > >
> > > State Map Generation: We first leverage the SSM's write recurrence from the SSM write equation to encode the entire 2D image feature map $F_0, ..., F_i$. This operation generates a compact, context-aware hidden state map, $S_{F_i}$.
> > >
> > > Deformable State Sampling: Queries $v_B$ attend to the state map $S_{F_i}$ through deformable sampling. Following BEVFormer, we lift the 2D BEV location (x, y) into a 3D pillar (x,y,z) and project Z evenly spaced pillar points onto each image for depth information. The resulting locations called reference points correspond to where an object at a BEV location would appear in the image based on the ego vehicle’s camera calibration and assuming there are no obstructions. Points inside image bounds form the valid reference set $\mathcal{E}\_{hit}$.
> > >
> > > We use $v_B$ to generate the offsets ${\Delta p}$ through a linear layer. We then sum the valid reference point of each camera view and the offset to obtain locations of sampling points $\mathcal{P}$. We then use bilinear interpolation to sample features from $S_{F_i}$ at $\mathcal{P}$. We fuse the sampled visual evidence back into the BEV query state along with weights $w$ learned by a linear layer through $S_B$. The specific process to get the updated BEV query hidden state $S_B$ is as follows:
> > >
> > > ${\cal P} = \mathcal{E}_{hit} + \Delta p$,
> > >
> > > $S_{B} =  \frac{1}{N}\sum_{n}^{N} w_n \cdot \phi(S_{F_i}, {\cal P}_n)$,
> > >
> > > where $\phi(S_{F_i}, \cal P_n)$ represents the use of bilinear interpolation function to extract the feature corresponding to position $\cal P$ on $S_{F_i}$. We refer the number of valid queries attended in a single-view map as: $|\mathcal{P}| = |\mathcal{E}_{hit}| \times |\Delta p|.$
> > >
> > > SSM Decode: The final output is generated from the enriched query state $S_B$ using the output projection equation:
> > > \begin{align}
> > > O &= S_B Q + VD,
> > > \end{align}
> > > where Q and D are the projection parameters associated with V.
> > >
> > >
> > >
> > >
> > > [1] Bevformer: Learning bird’s-eye-view representation from multi-camera images via spatiotemporal transformers.
> > >
> > > [2] Petrv2: A unified framework for 3d perception from multi-camera images
> > >
> > > [3] Mambev: Enabling state space models to learn birds-eye-view representations
> > >
> > > [4] Bevformerv2: Adapting modern image backbones to bird’s-eye-view recognition via
> > > perspective supervision.
> > >
> > > [5]  Feature pyramid networks for object detection.

---

### Official Review · Reviewer_tuTT · 2026-03-14

**Soundness:** 3
**Presentation:** 2
**Significance:** 4
**Originality:** 4
**Overall Recommendation:** 4
**Confidence:** 3

**Summary:**

This paper introduces Deformba, a vision state space model that integrates adaptive spatial sampling into the Mamba architecture. The core contribution is a Context-Aware State Fusion (CASF) module, which decouples the SSM into a write phase (producing a spatially structured state map) and a read phase (using learned offsets to sample relevant features from this map). This design aims to capture long-range dependencies while preserving spatial details and maintaining linear complexity. The method is evaluated on ImageNet classification, COCO detection/segmentation, ADE20K semantic segmentation, and nuScenes 3D detection, demonstrating competitive performance.

**Compliance With Llm Reviewing Policy:**

Affirmed.

**Key Questions For Authors:**

1. How does Deformba architecturally and conceptually differ from DefMamba (CVPR 2025)? Both methods introduce deformable operations into the Mamba framework. Could you provide a detailed comparison of the core mechanisms

2. Can you provide an ablation study that isolates the contribution of the deformable sampling mechanism?

3. What theoretical justification supports the use of gradient-based offset learning on the SSM state map? The paper motivates this design empirically but does not explain why the state map is a particularly suitable representation for deformable sampling, or how this approach fundamentally differs from deformable attention in Transformers.

4. Can you provide visualizations or statistical analyses of the learned sampling offsets? Understanding where the offsets sample, how they vary across layers and inputs, and whether they capture meaningful geometric deformations is crucial for interpreting the model's behavior.

**Limitations:**

Have you considered the potential limitations of Deformba under domain shift or with high-resolution inputs? For instance, could the learned offsets become unstable when the test distribution differs from training? Does state materialization pose a memory bottleneck for larger inputs?

**Strengths And Weaknesses:**

Strengths

1. The paper presents a well-engineered system that integrates SSMs with adaptive sampling. It conducts extensive experiments across four major vision tasks, covering multiple backbone scales. The comparisons span a wide range of modern architectures, including CNNs, ViTs, and recent SSMs, which strengthens the credibility of its empirical claims.

2. The idea of decoupling the SSM into write and read phases with deformable sampling is intuitively appealing for vision tasks, where spatial locality and adaptive receptive fields are important. The CASF module avoids the need for multiple scans (as required in bidirectional SSMs) and provides a flexible way to aggregate context.

3. The appendix includes thorough ablation studies on convolution causality, feedforward channels, expansion factor, number of attention heads, and BEV training settings. The discussion on I/O overhead and memory footprint demonstrates the authors' attention to practical deployment concerns, which is commendable.

Weakness

Major issue

1. The proposed Deformba is conceptually similar to the recently published DefMamba (CVPR 2025); both introduce deformable operations into the Mamba architecture to enhance visual modeling. However, the authors fail to provide a clear comparative analysis or elucidate the fundamental differences in mechanism design (e.g., offset generation, state handling) between Deformba and DefMamba. This omission makes it difficult to assess the novel contribution of this work.

2. Although the paper conducts ablation studies on convolutional causality and hyperparameters, it does not provide an ablation that isolates the effect of the deformable sampling mechanism itself. Is the performance improvement primarily attributable to the deformable mechanism, or does it stem from the increased parameter capacity? Furthermore, to what extent would replacing it with a simpler structure (e.g., an MLP) affect the results?

3. The motivation for employing deformable connections in SSMs is primarily empirical. The paper states that CASF "enhances adaptability and comprehensive scene understanding," but it provides no theoretical analysis to explain why gradient-based offset learning is particularly suitable for the SSM state map, nor how it fundamentally differs from deformable attention in Transformers.。

4. The CASF module essentially combines the "write-read decoupling" mechanism of SSMs with deformable sampling. A technique widely used in vision Transformers. The paper lacks a rigorous analysis of what new learning capabilities or theoretical insights this combination brings beyond mere engineering integration.

5. There is no analysis of how the learned offsets behave, where they sample, or whether they capture meaningful geometric deformations. No visualization or statistical analysis of the learned sampling points is provided, making it impossible to determine whether the model has truly learned adaptive spatial correspondences or to understand the behavioral patterns of these offsets across different layers and inputs.

Minor issue

1. The paper does not include a "Limitations" section or discuss potential issues in specific scenarios. For instance, could the learned offsets become unstable under domain shift? Discussing such points would improve the paper's completeness and aid future research in properly evaluating and using the method.

2. There is a noticeable language error ("and and") in Section 4.3.

---

> ### Author Rebuttal · Authors · 2026-03-31
>
> Dear Reviewer,
>
> Thank you for your thoughtful feedback! Please find our response to your concerns below:
>
> **W1 and Q1: Elucidate the fundamental differences Difference from DefMamba.**
>
> We thank the reviewer points out the comparison to DefMamba. We agree that both methods introduce deformable operations into visual SSMs, but the deformable operation is applied at fundamentally different stages. DefMamba makes the scan itself deformable that it predicts both 2D point offsets and a token-index offset, and thus dynamically changes the scanning order, which operates before the recurrent state is written. In contrast, our method, Deformba, keeps the SSM write pass fixed and introduces adaptivity only in the read stage. We first write the source tokens into the recurrent state through the standard SSM recurrence, $S_{t} = \alpha S_{t-1} + v_{t}k_{t}^{T}$, to generate state map $S_{2D}$. Offsets, $\mathcal{P}$, are then predicted from the written state map itself, and CASF performs adaptive bilinear sampling on the state map through $S_{Q} = \sum w \sigma(S_{2D}, \mathcal{P})$, $O=S_{Q}Q + VD$. Thus, our deformable operation is not a change of scan order, but an adaptive read over a written state memory.
>
>
> **W4: What new capability does the combination bring?**
>
> This distinction also leads to different functional scopes. DefMamba is designed to improve spatial modeling by changing how the state is written. Our CASF instead decouples write and read, so the same mechanism supports both self-attention-style mixing and cross-attention-style query-to-source retrieval with minimal changes to the write pass. We believe this write-once, adaptive-read-many paradigm is the key conceptual novelty of our work beyond simply adding a deformable module to Mamba.
>
> **W2 and Q2: Provide ablation to isolate the effect of the deformable sampling mechanism; replacing it with a simpler structure such as MLP.**
>
> We thank the reviewer for suggestion. Our current baseline in Table 6 already isolates the contribution of the deformable mechanism, as it uses the same SSM writing pipeline without CASF. The result shows that adding CASF improves Top-1 from 80.8 to 81.3/81.4 with only a small increase in parameters/FLOPs. It conducts a direct comparison between a non-deformable design and our full CASF module.
>
> We agree that this comparison does not fully isolate which part of CASF is responsible for the gain. To make this clearer, we add more targeted controls that we do not use ECA and MESA and keep the write-read framework unchanged but replace learned deformable sampling with predefined non-adaptive alternatives and MLP, such as:
>
> (i) zero-offset sampling, where each location only reads from its own reference point.
> (ii) fixed offset sampling, where we replace learned offsets with a predefined 1-hop local stencil around each reference point $\mathcal{E}(i, j)$
> (iii) MLP fusion, where we apply a MLP instead of sampling mechanism, in here, $\hat{S_{Q}} = W_{2} \sigma (W_{1}LN(S_{2D}))$, $\sigma$ is GELU.
>
> | Method | Im. size | Param. (M) | FLOPs (G)	 | Top-1 acc. |
> |:--|:--|:--:|:--:|:--:|
> |zero-offset| 224x224 | 25.0 | 4.7  | 80.8|
> |fixed offset|  224x224| 25.0 | 4.8  | 80.9|
> |MLP|  224x224| 30.8 | 5.6  | 81.0 |
> |dynamic offset|  224x224| 25.5 | 4.8 | 81.3|
>
> These show the improvement comes specifically from adaptive deformable state reading, rather than from increased capacity or from the existence of an extra fusion module alone.
>
>  **W3 and Q3: Theoretical analysis.**
>
> Thank you for your insightful comments. Our key point is not that deformable sampling itself is new, but that in SSMs it is more natural to place adaptivity in the read operator over the written state memory. The recurrent state is causal, while in vision, dependencies are inherently non-causal and in cross-query settings, the relevance of stored information is query-dependent. Hence, relying on the fixed write process alone would either bias interactions toward the scan order or require a much larger state to preserve all potentially evidence. CASF addresses this by keeping write pass compact and linear-time, but introducing adaptivity in the read operator.  The gradient-based offset learning is suitable here as the downstream task loss can directly optimize the read locations according to which memory slots are most useful for the current token. Deformable attention performs sparse sampling over raw key/value features, whereas CASF samples from a written SSM state memory after a fixed recurrent write pass. Therefore, CASF is better interpreted as adaptive memory retrieval rather than a direct reuse of deformable feature aggregation. We also provide the effective receptive field (ERF) of the offset prediction module with link below to further show the ERF difference.
>
> **W5 and Q4: Offset visualization.**
>
> We provide the visualization with link.
>
> [anonymous link](https://anonymous.4open.science/r/Deformba-Vision-State-Space-Model-with-Adaptive-State-Fusion-BAB4)

---

> > ### Author Rebuttal · Reviewer_tuTT · 2026-04-03
> >
> > I would like to thank the authors for their detailed and comprehensive rebuttal. Having carefully reviewed your responses, I find that my major issues have been addressed in a reasonable and convincing manner. The authors have clearly elucidated the fundamental mechanistic differences between Deformba and DefMamba, and the newly provided controlled ablation experiments are highly effective and well-executed.
> >
> > However, regarding the minor issue raised in my initial review, the authors did not explicitly address the request for an analysis of the model's limitations. This is an area I expect to see improved in the final version. Given the high-quality clarifications and additions provided during the rebuttal phase, particularly concerning the key mechanistic differences, ablation studies, and feature visualizations, I now have a much stronger appreciation for the architectural efficiency and academic contributions of this work. My primary concerns have been substantially resolved. I strongly recommend that the authors formally incorporate the aforementioned core comparative data, visual analyses, and the currently missing discussion on limitations into the main text or the appendix of the final manuscript. Thus, I will keep my rating.

---

> > > ### Author Response · Authors · 2026-04-05
> > >
> > > Thank you so much for your careful follow-up and positive reassessment of our rebuttal! We are glad that our clarifications regarding the mechanistic differences, controlled ablations, and visual analyses helped to address your main concerns.
> > >
> > > We appreciate your reminder regarding the minor issues. Because the rebuttal had a strict 5000 character limit, we first focused on the major technical questions and unfortunately did not have enough space to discuss those two minor issues. We would like to address it here.
> > >
> > > >**Minor issue 1: The paper does not include a "Limitations" section or discuss potential issues in specific scenarios. For instance, could the learned offsets become unstable under domain shift? Discussing such points would improve the paper's completeness and aid future research in properly evaluating and using the method.**
> > >
> > > Thank you for your suggestion! We agree our paper should include an explicit discussion about limitations. As you mentioned, a potential limitation is that the learned offsets are data-dependent and may become less reliable under challenging conditions, such as domain shift, large changes in object scale, cluttered backgrounds or atypical spatial layouts. Under such cases, CASF may sample regions that deviate from the key object evidence. This can reduce the effectiveness of adaptive state fusion. Moreover, Deformba should not be interpreted as only relying on CASF; as the architecture still benefits from additional locality priors including the convolutional components in each block. This suggests the gains come from the combination of compact state memory, adaptive readout, and local inductive bias rather than from a fully scan-agnostic spatial modeling mechanism.
> > >
> > > >**Minor issue 2: There is a noticeable language error ("and and") in Section 4.3.**
> > >
> > > Thank you for your careful reading of our paper and pointing this typo out! We will also correct the language issue in Section 4.3.
> > >
> > > Thank you again for your constructive feedback and for helping us improve the paper. We will make sure that the limitation section is in the final manuscript. And we will also incorporate the key comparative results, visual analyses and fixed writing typo from the rebuttal into paper.

---

### Decision · Program_Chairs · 2026-04-30

**Decision:**

Accept (regular)

**Comment:**

Dear Authors,

This draft received quite diverse ranking 3,3,4,5 (jyQX,, dSnq, tuTT, 3KcA). jyQX appreciated its solid experimental performance and empirical value and indicated it as a reason why not giving it a strong reject score, reviewer also stated it is not oppose to acceptance.

dSnq: "view the work as a solid empirical contribution with promising ideas. main concern relates to soundness and conceptual clarity. In particular, since the state map is constructed via prefix accumulation, it is not strictly spatially localized, and it remains somewhat unclear whether the subsequent sampling mechanism supports spatially meaningful retrieval in a principled sense. While the rebuttal improves clarity, this core point is not fully resolved."

jyQX, reviewer made few points pre requisite for the acceptance. "authors can commit to objectively acknowledging and discussing the theoretical limitations of the single scan (or providing a more reasonable explanation), and fully incorporating the missing core formulations into the main text in the camera-ready version"

Other two reviewers supported acceptance.

We strongly request to incorporate the raised points in the draft before making camera ready submission.

Congratulations.



regards

AC